# Towards Understanding the Condensation of Neural Networks at Initial Training

## Abstract

Implicit regularization is important for understanding the learning of neural networks (NNs). Empirical works show that input weights of hidden neurons (the input weight of a hidden neuron consists of the weight from its input layer to the hidden neuron and its bias term) condense on isolated orientations with a small initialization. The condensation dynamics implies that the training implicitly regularizes a NN towards one with much smaller effective size. In this work, we utilize multilayer networks to show that the maximal number of condensed orientations in the initial training stage is twice the multiplicity of the activation function, where "multiplicity" is multiple roots of activation function at origin. Our theoretical analysis confirms experiments for two cases, one is for the activation function of multiplicity one with arbitrary dimension input, which contains many common activation functions, and the other is for the layer with one-dimensional input and arbitrary multiplicity. This work makes a step towards understanding how small initialization implicitly leads NNs to condensation at initial training stage, which lays a foundation for the future study of the nonlinear dynamics of NNs and its implicit regularization effect at a later stage of training.

## 1 Introduction

Over-parameterized neural networks often show good generalization performance on real-world problems by minimizing loss functions without explicit regularization (Breiman, 1995; Zhang et al., 2017). For over-parameterized NNs, there are infinite possible sets of training parameters that can reach a satisfying training loss. However, their generalization performances can be very different. It is important to study what implicit regularization is imposed aside to the loss function during the training that leads the NN to a specific type of solutions.

Empirical works suggest that NNs may learn the data from simple to complex patterns (Arpit et al., 2017; Xu et al., 2019; Rahaman et al., 2019; Xu et al., 2020; Jin et al., 2020; Kalimeris et al., 2019). For example, an implicit bias of frequency principle is widely observed that NNs often learn the target function from low to high frequency (Xu et al., 2019; Rahaman et al., 2019; Xu et al., 2020), which has been utilized to understand various phenomena (Ma et al., 2020; Xu & Zhou, 2021) and inspiring algorithm design Liu et al. (2020). The NN output, either simple or complex, is a collective result of all neurons. The study of how neuron weights evolve during the training is central to understanding the collective behavior, including the complexity, of the NN output.

Luo et al. (2021) establish a phase diagram to study the effect of initialization on weight evolution for two-layer ReLU NNs at the infinite-width limit and find three distinct regimes in the phase diagram, i.e., linear regime, critical regime and condensed regime. The non-linear regime, a largely unexplored non-linear regime, is named as condensed regime because the input weights of hidden neurons (the input weight or the feature of a hidden neuron consists of the weight from its input layer to the hidden neuron and its bias term) condense on isolated orientations during the training (Luo et al., 2021). The three regimes are identified based on the relative change of input weights as the width approaches infinity, which tends to 0, $O(1)$ and $+\infty$, respectively.

The condensation is a feature learning process, which is important to the learning of DNNs. Note that in the following, **condensation is accompanied by a default assumption of small initialization or large relative change of input weights during training**. For practical networks, such as resnet18-like (He et al., 2016) in learning CIFAR10, as shown in Fig. 1(a) and Table 1, we find that

Table 1: Comparison of common (Glorot & Bengio, 2010) and condensed Gaussian initializations on resnet18. $\bar{m} = (m_{\text{in}} + m_{\text{out}})/2$. $m_{\text{in}}$: in-layer width. $m_{\text{out}}$: out-layer width.

| | common | | | condensed | | |
|---|---|---|---|---|---|---|
| | Glorot_uniform | Glorot_normal | $N(0, \frac{1}{\bar{m}})$ | $N(0, \frac{1}{m_{\text{out}}^4})$ | $N(0, \frac{1}{m_{\text{out}}^3})$ | $N(0, (\frac{1}{\bar{m}})^2)$ |
| Test 1 | 0.8807 | 0.8777 | 0.8816 | 0.8847 | 0.8824 | 0.8826 |
| Test 2 | 0.8857 | 0.8849 | 0.8806 | 0.8785 | 0.8813 | 0.8807 |
| Test 3 | 0.8809 | 0.8860 | 0.8761 | 0.8824 | **0.8861** | 0.8800 |

the performance of networks with initialization in the condensed regime is very similar to the common initialization methods. However, the condensation phenomenon provides an intuitive explanation of the good performance as follows, which may lead to a quantitative theoretical explanation in future work. The condensation transforms a large network to a network of only a few effective neurons, leading to an output function with low complexity. Since the complexity bounds the generalization error (Bartlett & Mendelson, 2002), the study of condensation could provide insight to how NNs are implicitly regularized to achieve good generalization performance in practice.

For two-layer ReLU NN, Maennel et al. (2018) prove that, as the initialization of parameters goes to zero, the features of hidden neurons condense at finite number of orientations depending on the input data; when performing a linearly separable classification task with infinite data, Pellegrini & Biroli (2020) show that at mean-field limit, a two-layer infinite-width ReLU NN is effectively equal to a NN of one hidden neuron, i.e., condensation on a single orientation. Both works (Maennel et al., 2018; Pellegrini & Biroli, 2020) study the condensation behavior for ReLU-NNs at an initial training stage in which the magnitudes of NN parameters are far smaller from well-fitting an $O(1)$ target function. However, it still remains unclear that **for NNs of more general activation functions, how the condensation emerges at the initial training stage.**

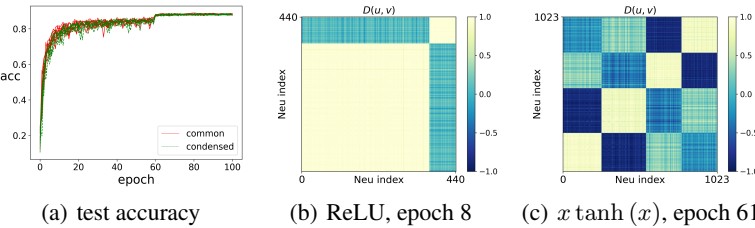

(a) test accuracy      (b) ReLU, epoch 8      (c) $x \tanh(x)$, epoch 61

Figure 1: The test accuracy in (a) and condensation in (b, c) of networks on CIFAR10. Each network consists of the convolution part of resnet18 and fully-connected (FC) layers with size 1024-1024-10 and softmax. The color in (b, c) indicates the inner product of normalized input weights of two neurons in the first FC layer, whose indexes are indicated by the abscissa and the ordinate. We discard about 55% of the hidden neurons, in which the $L_2$-norm of each input weight is smaller than 0.001, while remaining ones bigger than 0.05 in (b). The convolution part is equipped with ReLU activation and initialized by Glorot normal distribution (Glorot & Bengio, 2010). For FC layers in (a), the activation is ReLU and they are initialized by three common methods (red) and three condensed ones (green) as indicated in Table 1. The learning rate is $10^{-3}$ for epoch 1-60 and $10^{-4}$ for epoch 61-100. For (b, c), the learning rate is $5 \times 10^{-6}$ for visualization and FC layers are initialized by $N(0, \frac{1}{m_{\text{out}}^3})$ and equipped with ReLU in (b) and $x \tanh(x)$ in (c) as activation functions. Adam optimizer with cross-entropy loss and batch size 128 are used for all experiments.

In this work, we show that the condensation at the initial stage is closely related to the multiplicity $p$ at $x = 0$, which means the derivative of activation at $x = 0$ is zero up to the $(p-1)th$-order and is non-zero for the $p$-th order. To verify their relation, we use the common activation function $\text{sigmoid}(x)$, $\text{softplus}(x)$, $\tanh(x)$, which are multiplicity one, and variants of $\tanh(x)$,i.e. $x \tanh(x)$ and $x^2 \tanh(x)$ with multiplicity two and three, for our experiments. For comparison, we also show the initial condensation of $\text{ReLU}(x)$, which is studied previously (Maennel et al., 2018) and has totally different properties at origin compared with $\tanh(x)$. Our experiments suggest that

the maximal number of condensed orientations is twice the multiplicity of the activation function used in general NNs. For finite-width two-layer NNs with small initialization at the initial training stage, each hidden neuron's output in a finite domain around 0 can be approximated by a $p$-th order polynomial and so is the NN output function. Based on the $p$-th order approximation, we show a preliminary theoretical support for condensation by a theoretical analysis for two cases, one is for the activation function of multiplicity one with arbitrary dimension input, which contains many common activation functions, and the other is for the layer with one-dimensional input and arbitrary multiplicity. Therefore, small initialization imposes an implicit regularization that restricts the NN to be effectively much narrower neural network at the initial training stage. As commonly used activation functions, such as $\tanh(x)$, $\mathrm{sigmoid}(x)$, $\mathrm{softplus}(x)$, etc., are all multiplicity one, our study of initial training behavior lays an important basis for the further study of implicit regularization throughout the training.

## 2 RELATED WORKS

A research line studies how initialization affects the weight evolution of NNs with a sufficiently large or infinite width. For example, with an initialization in the neural tangent kernel (NTK) regime or lazy training regime (weights change slightly during the training), the gradient flow of infinite-width NN, can be approximated by a linear dynamics of random feature model (Jacot et al., 2018; Arora et al., 2019; Zhang et al., 2020; E et al., 2020; Chizat & Bach, 2019), whereas for the initialization in the mean-field regime (weights change significantly during the training), the gradient flow of infinite-width NN exhibits highly nonlinear dynamics (Mei et al., 2019; Rotskoff & Vanden-Eijnden, 2018; Chizat & Bach, 2018; Sirignano & Spiliopoulos, 2020). Pellegrini & Biroli (2020) analyze how the dynamics of each parameter transforms from a lazy regime (NTK initialization) to a rich regime (mean-field initialization) for an two-layer infinite-width ReLU NN to perform a linearly separable classification task with infinite data. Luo et al. (2021) systematically study the effect of initialization for two-layer ReLU NN with infinite width by establishing a phase diagram, which shows three distinct regimes, i.e., linear regime (similar to the lazy regime), critical regime and condensed regime (similar to the rich regime), based on the relative change of input weights as the width approaches infinity, which tends to 0, $O(1)$ and $+\infty$, respectively. Luo et al. (2021) also empirically find that, in the condensed regime, the features of hidden neurons (orientation of the input weight) condense in several isolated orientations, which is a strong feature learning behavior, an important characteristic of deep learning, however, in Luo et al. (2021), it is not clear how general of the condensation when other activation functions are used and why there is condensation.

## 3 PRELIMINARY: NEURAL NETWORKS AND INITIAL STAGE

A two-layer NN is

$$f_{\boldsymbol{\theta}}(\boldsymbol{x}) = \sum_{j=1}^{m} a_j \sigma(\boldsymbol{w}_j \cdot \boldsymbol{x}), \tag{1}$$

where $\sigma(\cdot)$ is the activation function, $\boldsymbol{w}_j = (\bar{\boldsymbol{w}}_j, \boldsymbol{b}_j) \in \mathbb{R}^{d+1}$ is the neuron feature including the input weight and bias terms, and $\boldsymbol{x} = (\bar{\boldsymbol{x}}, 1) \in \mathbb{R}^{d+1}$ is combination of the input sample and scalar 1, $\boldsymbol{\theta}$ is the set of all parameters, i.e., $\{a_j, \boldsymbol{w}_j\}_{j=1}^{m}$. For simplicity, **we call $\boldsymbol{w}_j$ as input weight or weight** and $\boldsymbol{x}$ as input sample.

A $L$-layer NN can be recursively defined by feeding the output of the previous layer as the input to the current hidden layer i.e.,

$$\boldsymbol{x}^{[0]} = (\boldsymbol{x}, 1), \quad \boldsymbol{x}^{[1]} = (\sigma(\boldsymbol{W}^{[1]}\boldsymbol{x}^{[0]}), 1), \quad \boldsymbol{x}^{[l]} = (\sigma(\boldsymbol{W}^{[l]}\boldsymbol{x}^{[l-1]}), 1), \text{ for } l \in \{2, 3, ..., L\}$$
$$f(\boldsymbol{\theta}, \boldsymbol{x}) = \frac{1}{\alpha}\mathbf{a}^{\mathsf{T}}\boldsymbol{x}^{[L]} \triangleq f_{\boldsymbol{\theta}}(\boldsymbol{x}), \tag{2}$$

where $\boldsymbol{W}^{[l]} = (\bar{\boldsymbol{W}}^{[l]}, \boldsymbol{b}^{[l]}) \in \mathbb{R}^{m_l \times (m_{l-1}+1)}$, and $m_l$ represents the dimension of the $l$-th hidden layer. For simplicity, **we also call each row of $\boldsymbol{W}^{[l]}$ as input weight or weight** and $\boldsymbol{x}^{[l-1]}$ as input

neurons. The target function is denoted as $f^*(\boldsymbol{x})$. The training loss function is mean squared error

$$R_S(\boldsymbol{\theta}) = \frac{1}{2n} \sum_{i=1}^{n} (f_{\boldsymbol{\theta}}(\boldsymbol{x}_i) - f^*(\boldsymbol{x}_i))^2. \tag{3}$$

Without loss of generality, we assume that the output is one-dimensional for theoretical analysis, because for high-dimensional cases, we only need to sum the components directly. For summation, it does not affect the results of our theories. We consider the gradient flow training

$$\dot{\boldsymbol{\theta}} = -\nabla_{\boldsymbol{\theta}} R_S(\boldsymbol{\theta}). \tag{4}$$

For convenience, we characterize the activation function by the following definition.

**Definition 1** (multiplicity $p$). *Suppose that $\sigma(x)$ satisfies the following condition, there exists a $p \in \mathbb{N}$ and $p \geq 1$, such that the k-th order derivative $\sigma^{(k)}(0) = 0$ for $k = 1, 2, \cdots, p-1$, and $\sigma^{(p)}(0) \neq 0$, then we say $\sigma$ has multiplicity $p$.*

In the experiments, we study the condensation at the initial stage of training. For a fixed loss, the step we need to achieve it is highly related to the size of learning rate. Therefore, we propose a definition of the initial stage of training by the size of loss in this article, that is the stage before the value of loss function decays to 70% of its initial value. Such a definition is reasonable, for generally a loss could decay to 1% of its initial value or even lower. The loss of the all experiments in the article can be found in Appendix A.3, and they do meet the definition of the initial stage here.

## 4 INITIAL CONDENSATION OF INPUT WEIGHTS

It is intuitively believed that NNs are powerful at learning data features, which should be an important reason behind the success of deep learning. A simple way to define a learned feature of a neuron is by the orientation of its input weights. Previous work in Luo et al. (2021) show that there is a condensed regime, where the neuron features condense on isolated orientations during the training for two-layer ReLU NNs. The condensation implies that although there are many more neurons than samples, the number of effective neurons, i.e., the number of different used features in fitting, is often much smaller than the number of samples. Therefore, the condensation provides a potential mechanism that helps over-parameterized NNs avoid overfitting. However, it is still unclear how the condensation, for general NNs with small initialization, emerges during the training. In this section, we would empirically show how the condensation differs among NNs with activation functions of different multiplicities, followed by theoretical analysis in the next section.

### 4.1 EXPERIMENTAL SETUP

For Synthetic dataset and MNIST: Throughout this work, we use fully-connected neural network with size, $d$-$m$-$\cdots$-$m$-$d_{out}$. The input dimension $d$ is determined by the training data. The output dimension is $d_{out} = 1$ for synthetic data and $d_{out} = 10$ for MNIST. The number of hidden neurons $m$ is specified in each experiment. All parameters are initialized by a Gaussian distribution $N(0, var)$. The total data size is $n$. The training method is Adam with full batch, learning rate $lr$ and MSE loss. For synthetic data, we sample the training data uniformly from a sub-domain of $\mathbb{R}^d$.

For CIFAR10 and CIFAR100 dataset: We use Resnet18-like neural network, which has been describe in Fig. 1 thoroughly. The input dimension $d$ is determined by the training data. The output dimension is $d_{out} = 10$ for CIFAR10 and $d_{out} = 100$ for CIFAR100. All parameters are initialized by a Gaussian distribution $N(0, var)$. The total data size is $n$. The training method is Adam with batch size 128, learning rate $lr$ and Cross-entropy loss.

### 4.2 MULTIDIMENSIONAL DATA

We first show the condensation at initial training stage in fitting multidimensional dataset. Since the input is a multidimensional vector, the direction is also multidimensional. To characterize the condensation, we use $D(\boldsymbol{u}, \boldsymbol{v})$ to denote the inner product of the normalized vectors of two input weights, i.e., $D(\boldsymbol{u}, \boldsymbol{v}) = \boldsymbol{u}^{\mathsf{T}} \boldsymbol{v}$.

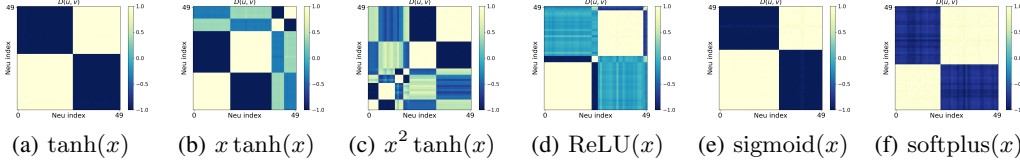

(a) $\tanh(x)$    (b) $x\tanh(x)$    (c) $x^2\tanh(x)$    (d) $\mathrm{ReLU}(x)$    (e) $\mathrm{sigmoid}(x)$    (f) $\mathrm{softplus}(x)$

Figure 2: Condensation of two-layer NNs. The color indicates $D(u,v)$ of two hidden neurons' input weights at epoch 100, whose indexes are indicated by the abscissa and the ordinate, respectively. If neurons are in the same beige block, $D(u,v) \sim 1$ (navy-blue block, $D(u,v) \sim -1$), their input weights have the same (opposite) direction. The activation functions are indicated by the sub-captions. The training data is 80 points sampled from $\sum_{k=1}^{5} 3.5\sin(5x_k + 1)$, where each $x_k$ is uniformly sampled from $[-4, 2]$. $n = 80$, $d = 5$, $m = 50$, $d_{out} = 1$, $var = 0.005^2$. $lr = 10^{-3}, 8 \times 10^{-4}, 2.5 \times 10^{-4}$ for (a-d), (e) and (f), respectively.

We use a two-layer fully-connected NN with size 5-50-1 to fit $n = 80$ training data sampled from a 5-dimensional function $\sum_{k=1}^{5} 3.5\sin(5x_k + 1)$, where $\boldsymbol{x} = (x_1, x_2, \cdots, x_5)^\mathsf{T} \in \mathbb{R}^5$ and each $x_k$ is uniformly sampled from $[-4, 2]$. As shown in Fig. 2(a), for activation function $\tanh(x)$, the color indicates $D(u,v)$ of two hidden neurons' weights at epoch 100, whose indexes are indicated by the abscissa and the ordinate. If the neurons are in the same beige block, $D(u,v) \sim 1$ (navy-blue block, $D(u,v) \sim -1$), their input weights have the same (opposite) direction. Obviously, input weights of hidden neurons condense at two opposite directions, i.e., one line. As the multiplicity increasing, NNs with $x\tanh(x)$ (Fig. 2(b)) and $x^2\tanh x$ (Fig. 2(c)) condense at two and three different lines, respectively. For activation function $\mathrm{sigmoid}(x)$ in Fig. 2(d) and $\mathrm{softplus}(x)$ in Fig. 2(e), which are frequently used and have multiplicity one, NNs also condense at two opposite directions. For ReLU in Fig. 2(f), for which the multiplicity definition cannot apply, the NN condenses at three directions, in which two are opposite. Through these experiments, we conjecture that the maximal number of condensed orientations is twice the multiplicity of the activation function used at initial training.

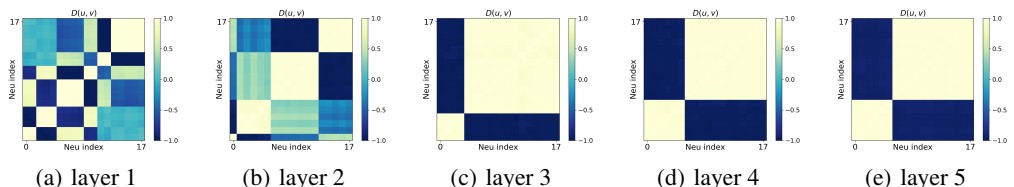

(a) layer 1      (b) layer 2      (c) layer 3      (d) layer 4      (e) layer 5

Figure 3: Condensation of six-layer NNs with residual connections. The activation functions for hidden layer 1 to hidden layer 5 are $x^2\tanh(x)$, $x\tanh(x)$, $\mathrm{sigmoid}(x)$, $\tanh(x)$ and $\mathrm{softplus}(x)$, respectively. The numbers of steps selected in the sub-pictures are epoch 1000, epoch 900, epoch 900, epoch 1400 and epoch 1400, respectively, while the NN is only trained once. The color indicates $D(u,v)$ of two hidden neurons' input weights, whose indexes are indicated by the abscissa and the ordinate, respectively. The training data is 80 points sampled from a 3-dimensional function $\sum_{k=1}^{3} 4\sin(12x_k + 1)$, where each $x_k$ is uniformly sampled from $[-4, 2]$. $n = 80$, $d = 3$, $m = 18$, $d_{out} = 1$, $var = 0.01^2$, $lr = 4 \times 10^{-5}$.

For multilayer NNs with different activation functions, we show that the condensation for all hidden layers is similar to the two-layer NNs. In deep networks, residual connection is often introduced to overcome the vanishing of gradient. To show the generality of condensation, we perform an experiment of six-layer NNs with residual connections. To show the difference of various activation functions, we set the activation functions for hidden layer 1 to hidden layer 5 as $x^2\tanh(x)$, $x\tanh(x)$, $\mathrm{sigmoid}(x)$, $\tanh(x)$ and $\mathrm{softplus}(x)$, respectively. The structure of the residual is $\boldsymbol{h}_{l+1}(\boldsymbol{x}) = \sigma(\boldsymbol{W}_l\boldsymbol{h}_l(\boldsymbol{x}) + b_l) + \boldsymbol{h}_l(\boldsymbol{x})$, where $\boldsymbol{h}_l(\boldsymbol{x})$ is the output of the $l$-th layer. As shown in Fig. 3, input weights condense at three, two, one, one and one lines for hidden layer 1 to hidden layer 5, respectively. Note that residual connections are not necessary. We show an experiment of the same structure as in Fig. 3 but without residual connections in Appendix A.6. To show the universality of condensation, we train resnet18-like neural networks to learn CIFAR10. We study

the condensation of the first fully connected layer of the network, using ReLU and $x \tanh(x)$ as the activation functions and initialization distribution $N(0, (\frac{1}{m^{1.5}})^2)$. As shown in Fig. 1 (b) and (c), the condensations for activation $\text{ReLU}(x)$ and $x \tanh(x)$ are consistent with Fig. 2 and our conjecture. More experiments on dataset CIFAR10 and CIFAR100 can be found in Appendix A.5.

We also find that when the training data is less oscillated, the NN may condense at fewer directions. For example, as shown in Fig. 4(a), compared with the high frequency function in Fig. 2, we only change the target function to be a lower-frequency function, i.e., $\sum_{k=1}^{5} 3.5 \sin(2x_k + 1)$. In this case, the NN with $x^2 \tanh(x)$ only condenses at three directions, in which two are opposite. For MNIST data in Fig. 4(b), we find that, the NN with $x^2 \tanh(x)$ condenses at one line, which may suggest that the function for fitting MNIST dataset is a low-frequency function. For CIFAR100 data in Fig. 4(c), we find that input weights of the first FC layer with $x \tanh(x)$ condense at only one line, which implies that features extracted by the convolution part of the NN may own low complexity.

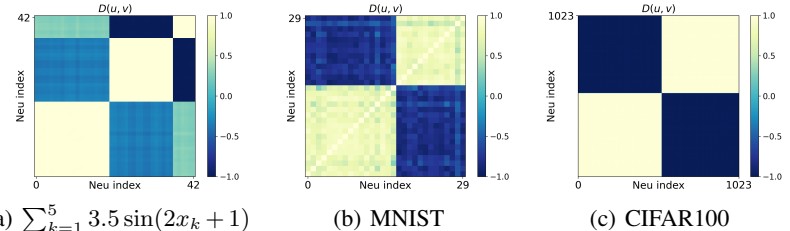

(a) $\sum_{k=1}^{5} 3.5 \sin(2x_k + 1)$     (b) MNIST     (c) CIFAR100

Figure 4: Condensation of low-frequency functions with two-layer NNs in (a,b) and condensation of the first FC layer of the Resnet18-like network on CIFAR100 in (c). The color indicates $D(u, v)$ of two hidden neurons' input weights, whose indexes are indicated by the abscissa and the ordinate. For (a,b), two-layer NN at epoch: 100 with activation function: $x^2 \tanh(x)$. For (a), we discard about 15% of hidden neurons, in which the $L_2$-norm of each input weight is smaller than 0.04, while remaining those bigger than 0.4. The mean magnitude here for each parameter is $(0.4^2/785)^{0.5}$ $\sim 0.01$, which should also be quite small. All settings in (a) are the same as Fig. 2, except for the lower frequency target function. Parameters for (b) are $n = 60000$, $d = 784$, $m = 30$, $d_{out} = 10$, $var = 0.001^2$. $lr = 5 \times 10^{-5}$. The structure and parameters of the Resnet18-like neural network for (c) is the same as Fig. 1(b), except for the data set CIFAR100 and learning rate $lr = 1 \times 10^{-6}$.

To understand the mechanism of the initial condensation, we turn to experiments of 1-d input and two-layer NN, which can be clearly visualized in the next subsection.

### 4.3 1-D INPUT AND TWO-LAYER NN

For 1-d data, we visualize the evolution of the two-layer NN output and each weight, which confirms the connection between the condensation and the multiplicity of the activation function.

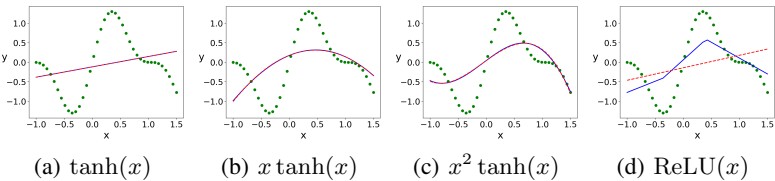

(a) $\tanh(x)$     (b) $x \tanh(x)$     (c) $x^2 \tanh(x)$     (d) $\text{ReLU}(x)$

Figure 5: The outputs of two-layer NNs at epoch 1000 with activation functions $\tanh(x)$, $x \tanh(x)$, $x^2 \tanh(x)$, and $\text{ReLU}(x)$ are displayed, respectively. The training data is 40 points uniformly sampled from $\sin(3x) + \sin(6x)/2$ with $x \in [-1, 1.5]$, illustrated by green dots. The blue solid lines are the NN outputs at test points, while the red dashed auxiliary lines are the first, second, third and first order polynomial fittings of the test points for (a, b, c, d), respectively. Parameters are $n = 40$, $d = 1$, $m = 100$, $d_{out} = 1$, $var = 0.005^2$, $lr = 5 \times 10^{-4}$.

We display the outputs at initial training, epoch 1000, in Fig. 5. Due to the small magnitude of parameters, an activation function with multiplicity $p$ can be well approximated by a $p$-th order polynominal, thus, the NN output can also be approximated by a $p$-th order polynominal. As shown in Fig. 5(a-c), the NN outputs with activation functions $\tanh(x)$, $x\tanh(x)$ and $x^2\tanh(x)$ overlap well with the auxiliary of a linear, a quadratic and a cubic polynominal curve, respectively. In Fig. 5(d), the NN output with ReLU activation function deviates from a linear function (red auxiliary line). Particularly, the NN output has several sharp turning points. This experiment, although simple, but convincingly shows that NN does not always learn a linear function at the initial training stage and the complexity of such learning depends on the activation function.

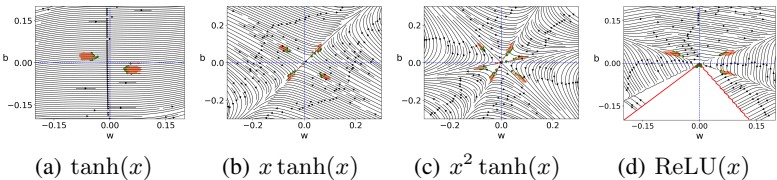

(a) $\tanh(x)$     (b) $x\tanh(x)$     (c) $x^2\tanh(x)$     (d) $\mathrm{ReLU}(x)$

Figure 6: The direction field for input weight $\boldsymbol{w} := (w, b)$ of the dynamics in (4.3) at epoch 200. All settings are the same as Fig. 5. Around the original point, the field has one, two, three stables lines, on which an input weight would keep its direction, for $\tanh(x)$, $x\tanh(x)$, and $x^2\tanh(x)$, respectively. We also display the value of each weight by the green dots and the corresponding directions by the orange arrows.

We visualize the direction field for input weight $\boldsymbol{w}_j := (w_j, b_j)$, following the gradient flow,

$$\dot{\boldsymbol{w}}_j = -\frac{a_j}{n}\sum_{i=1}^{n} e_i \sigma'(\boldsymbol{w}_j \cdot \boldsymbol{x}_i)\boldsymbol{x}_i,$$

where $e_i := f_{\boldsymbol{\theta}}(\boldsymbol{x}_i) - f^*(\boldsymbol{x}_i)$. Since we only care about the direction of $\boldsymbol{w}_j$ and $a_j$ is a scalar at each epoch, we can visualize $\dot{\boldsymbol{w}}_j$ by $\dot{\boldsymbol{w}}_j/a_j$. For simplicity, we do not distinguish $\dot{\boldsymbol{w}}_j/a_j$ and $\dot{\boldsymbol{w}}_j$ if there is no ambiguity. When we compute $\dot{\boldsymbol{w}}_j$ for different $j$'s, $e_i \boldsymbol{x}_i$ for $(i = 1, \cdots, n)$ is independent with $j$. Then, at each epoch, for a set of $\{e_i, \boldsymbol{x}_i\}_{i=1}^{n}$, we can consider the following direction field

$$\dot{\boldsymbol{\omega}} = -\frac{1}{n}\sum_{i=1}^{n} e_i \boldsymbol{x}_i \sigma'(\boldsymbol{\omega} \cdot \boldsymbol{x}_i).$$

When $\boldsymbol{\omega}$ is set as $\boldsymbol{w}_j$, we can obtain $\dot{\boldsymbol{w}}_j$. As shown in Fig. 6, around the original point, the field has one, two, three stables lines, on which a neuron would keep its direction, for $\tanh(x)$, $x\tanh(x)$, and $x^2\tanh(x)$, respectively. We also display the input weight of each neuron on the field by the green dots and their corresponding velocity directions by the orange arrows. Similarly to the high-dimensional cases, NNs with multiplicity $p$ activation functions condense at $p$ different lines for $p = 1, 2, 3$. Therefore, It is reasonable to conjecture that the maximal number of condensed orientations is twice the multiplicity of the activation function used. As shown in Fig. 6(d), the field and the condensation for the NN with $\mathrm{ReLU}(x)$ is much more complex.

Taken together, we have empirically shown that the multiplicity of the activation function is a key factor that determines the complexity of the initial output and condensation. In figures of this section, we only show the results of the final step of condensation. To facilitate the understanding of evolution of condensation in the initial stage, we show several steps during the initial stage of each example in Appendix A.4. From the evolution, we could directly observe how condensation occurs.

## 5    ANALYSIS OF THE INITIAL CONDENSATION OF INPUT WEIGHTS

In this section, we would present a preliminary analysis to understand how the multiplicity of the activation function affects the initial condensation. At each training step, we consider the velocity field of weights in each hidden layer of a neural networks.

Considering a network with $L$ hidden layers, we use row vector $\boldsymbol{W}_j^{[k]}$ to represent the weight from the $(k\text{-}1)$-th layer to the $j$-th neuron in the $k$-th layer. For each $k$ and $j$, $\boldsymbol{W}_j^{[k]}$ satisfies the following

dynamics, (see Appendix A.2)

$$\dot{r} = \boldsymbol{u} \cdot \dot{\boldsymbol{w}}, \quad \dot{\boldsymbol{u}} = \frac{\dot{\boldsymbol{w}} - (\dot{\boldsymbol{w}} \cdot \boldsymbol{u})\boldsymbol{u}}{r}. \tag{5}$$

where $\boldsymbol{w}$ can be $\boldsymbol{W}_j^{[k]\mathsf{T}}$ for all $k$'s and $j$'s, $r = \|\boldsymbol{w}\|_2$ is the amplitude, and $\boldsymbol{u} = \boldsymbol{w}/r$.

Suppose the activation function has multiplicity $p$, i.e., $\sigma^{(k)}(0) = 0$ for $k = 1, 2, \cdots, p - 1$, and $\sigma^{(p)}(0) \neq 0$. For convenience, we define an operator $\mathcal{P}$ satisfying $\mathcal{P}\boldsymbol{w} := \dot{\boldsymbol{w}} - \boldsymbol{u}(\dot{\boldsymbol{w}} \cdot \boldsymbol{u})$. Condensation refers to that the weight evolves towards a direction that will not change in the direction field and is defined as follows,

$$\dot{\boldsymbol{u}} = 0 \iff \mathcal{P}\boldsymbol{w} := \dot{\boldsymbol{w}} - \boldsymbol{u}(\dot{\boldsymbol{w}} \cdot \boldsymbol{u}) = 0.$$

Since $\dot{\boldsymbol{w}} \cdot \boldsymbol{u}$ is a scalar, $\dot{\boldsymbol{w}}$ is parallel with $\boldsymbol{u}$. $\boldsymbol{u}$ is a unit vector, therefore, we have $\boldsymbol{u} = \dot{\boldsymbol{w}}/\|\dot{\boldsymbol{w}}\|_2$. In this work, we consider NNs with sufficiently small parameters. For example, suppose $r = \|\boldsymbol{w}\|_2 \sim O(\epsilon)$, $\epsilon$ is a small quantity. Dynamics (5) shows that $O(\dot{r}) \sim O(\dot{\boldsymbol{w}})$ and $O(\dot{\boldsymbol{u}}) \sim O(\dot{r})/O(\epsilon)$. Therefore, the orientation $\boldsymbol{u}$ would moves much more quickly than the amplitude $r$. In the following, we study the case of (i) $p = 1$ and (ii) $m_l = 1$.

## 5.1 CASE 1: $p = 1$

Since we have (see Appendix A.2),

$$\boldsymbol{w}^{\mathsf{T}} = \dot{\boldsymbol{W}}_j^{[k]} = -\frac{1}{n} \sum_{i=1}^{n} (f(\boldsymbol{\theta}, \boldsymbol{x}_i) - y_i) \left[ \operatorname{diag}\{\sigma'(\boldsymbol{W}^{[k]} \boldsymbol{x}_i^{[k-1]})\} (E^{[k+1:L]} \boldsymbol{a}) \right]_j \boldsymbol{x}_i^{[k-1]\mathsf{T}},$$

where we use $E^l = \boldsymbol{W}^{[l]\mathsf{T}} \operatorname{diag}\{\sigma'(\boldsymbol{W}^{[l]} \boldsymbol{x}^{[l-1]})\}$, for $l \in \{2, 3, ..., L\}$, $E^{[q:p]} = E^q E^{q+1} ... E^p$.

For a fixed step, we only consider the gradient w.r.t. $\dot{\boldsymbol{W}}_j^{[k]}$. Suppose $\sigma'(0) \neq 0$ and parameters are small. Denote $e_i := (f(\boldsymbol{\theta}, \boldsymbol{x}_i) - y_i)$. By Taylor expansion,

$$\mathcal{P}\boldsymbol{w} \overset{\text{leading order}}{\approx} \mathcal{Q}\boldsymbol{w} := -\frac{1}{n} (\operatorname{diag}\{\sigma'(\boldsymbol{0})\} (E^{[k+1:L]} \boldsymbol{a}))_j \sum_{i=1}^{n} e_i \boldsymbol{x}_i^{[k-1]}$$

$$+ \left( \frac{1}{n} (\operatorname{diag}\{\sigma'(\boldsymbol{0})\} (E^{[k+1:L]} \boldsymbol{a}))_j \sum_{i=1}^{n} e_i \boldsymbol{x}_i^{[k-1]} \cdot \boldsymbol{u} \right) \boldsymbol{u} = 0,$$

where operator $\mathcal{Q}$ is the leading-order approximation of operator $\mathcal{P}$, and here $E^{[k+1:L]}$ is independent with $i$ because $\operatorname{diag}\{\sigma'(\boldsymbol{W}^{[l]} \boldsymbol{x}^{[l-1]})\} \approx \operatorname{diag}\{\sigma'(\boldsymbol{0})\}$. Since $\operatorname{diag}\{\sigma'(\boldsymbol{0})\} = \boldsymbol{I}$, and, WLOG, we assume $a \neq 0$, then

$$\mathcal{Q}\boldsymbol{w} = 0 \iff \sum_{i=1}^{n} e_i \boldsymbol{x}_i^{[k-1]} = \left( \sum_{i=1}^{n} e_i \boldsymbol{x}_i^{[k-1]} \cdot \boldsymbol{u} \right) \boldsymbol{u}.$$

We have

$$\boldsymbol{u} = \frac{\sum_{i=1}^{n} e_i \boldsymbol{x}_i^{[k-1]}}{\|\sum_{i=1}^{n} e_i \boldsymbol{x}_i^{[k-1]}\|_2} \quad or \quad \boldsymbol{u} = -\frac{\sum_{i=1}^{n} e_i \boldsymbol{x}_i^{[k-1]}}{\|\sum_{i=1}^{n} e_i \boldsymbol{x}_i^{[k-1]}\|_2}.$$

This calculation shows that for layer $k$, the input weights for any hidden neuron $j$ have the same two stable directions. Therefore, when parameters are sufficiently small, which implies that the orientation $\boldsymbol{u}$ would moves much more quickly than the amplitude $r$, all input weights towards converging to the same direction or the opposite direction, i.e., condensation on a line.

## 5.2 CASE 2: $m_l = 1$

By the definition of the multiplicity $p$, we have

$$\sigma'(\boldsymbol{w} \cdot \boldsymbol{x}_i) = \frac{\sigma^{(p)}(0)}{(p-1)!} (\boldsymbol{w} \cdot \boldsymbol{x}_i)^{p-1} + o((\boldsymbol{w} \cdot \boldsymbol{x}_i)^{p-1}).$$

where $(\cdot)^{p-1}$ and $\sigma^p(\cdot)$ operate on component here. Then up to the leading order in terms of the magnitude of $\boldsymbol{\theta}$, we have (see Appendix A.2)

$$\mathcal{P}\boldsymbol{w} \stackrel{\text{leading order}}{\approx} \mathcal{Q}\boldsymbol{w} := -(\frac{1}{n}\sum_{i=1}^{n} e_i \boldsymbol{x}_i^{[k-1]}(\boldsymbol{w}^\intercal \boldsymbol{x}_i^{[k-1]})^{p-1})[\text{diag}\{\frac{\sigma^{(p)}(\boldsymbol{0})}{(p-1)!}\}(E^{[k+1:L]}\mathbf{a})]_j$$

$$+((\frac{1}{n}\sum_{i=1}^{n} e_i \boldsymbol{x}_i^{[k-1]}(\boldsymbol{w}^\intercal \boldsymbol{x}_i^{[k-1]})^{p-1})[\text{diag}\{\frac{\sigma^{(p)}(\boldsymbol{0})}{(p-1)!}\}(E^{[k+1:L]}\mathbf{a})]_j \cdot \boldsymbol{u})\boldsymbol{u}.$$

WLOG, we also assume $a \neq 0$. And by definition, $\boldsymbol{w} = r\boldsymbol{u}$, we have

$$\mathcal{Q}\boldsymbol{w} = 0 \Leftrightarrow \boldsymbol{u} = \frac{\frac{1}{n}\sum_{i=1}^{n} e_i \boldsymbol{x}_i^{[k-1]}(\boldsymbol{u}^\intercal \boldsymbol{x}_i^{[k-1]})^{p-1}}{\|\frac{1}{n}\sum_{i=1}^{n} e_i \boldsymbol{x}_i^{[k-1]}(\boldsymbol{u}^\intercal \boldsymbol{x}_i^{[k-1]})^{p-1}\|_2}$$

$$or \ \boldsymbol{u} = -\frac{\frac{1}{n}\sum_{i=1}^{n} e_i \boldsymbol{x}_i^{[k-1]}(\boldsymbol{u}^\intercal \boldsymbol{x}_i^{[k-1]})^{p-1}}{\|\frac{1}{n}\sum_{i=1}^{n} e_i \boldsymbol{x}_i^{[k-1]}(\boldsymbol{u}^\intercal \boldsymbol{x}_i^{[k-1]})^{p-1}\|_2}.$$

Since $d + 1 = 2$, we denote $\boldsymbol{u} = (u_1, u_2)^\intercal \in \mathbb{R}^2$ and $\boldsymbol{x}_i^{[k-1]} = ((\boldsymbol{x}_i^{[k-1]})_1, (\boldsymbol{x}_i^{[k-1]})_2)^\intercal \in \mathbb{R}^2$, then,

$$\frac{\sum_{i=1}^{n}(u_1(\boldsymbol{x}_i^{[k-1]})_1 + u_2(\boldsymbol{x}_i^{[k-1]})_2)^{p-1}e_i(\boldsymbol{x}_i^{[k-1]})_1}{\sum_{i=1}^{n}(u_1(\boldsymbol{x}_i^{[k-1]})_1 + u_2(\boldsymbol{x}_i^{[k-1]})_2)^{p-1}e_i(\boldsymbol{x}_i^{[k-1]})_2} = \frac{u_1}{u_2} \triangleq \hat{u}.$$

We obtain the equation for $\hat{u}$,

$$\sum_{i=1}^{n}(\hat{u}(\boldsymbol{x}_i^{[k-1]})_1 + (\boldsymbol{x}_i^{[k-1]})_2)^{p-1}e_i(\boldsymbol{x}_i^{[k-1]})_1 = \hat{u}\sum_{i=1}^{n}(\hat{u}(\boldsymbol{x}_i^{[k-1]})_1 + (\boldsymbol{x}_i^{[k-1]})_2)^{p-1}e_i(\boldsymbol{x}_i^{[k-1]})_2.$$

Since it is an univariate $p$-th order equation, $\hat{u} = \frac{u_1}{u_2}$ has at most $p$ complex roots. Because $\boldsymbol{u}$ is a unit vector, $\boldsymbol{u}$ at most has $p$ pairs of values, in which each pair are opposite.

Taken together, our theoretical analysis is consistent with our experiments, that is, the maximal number of condensed orientations is twice the multiplicity of the activation function used when parameters are small. Besides, because commonly used activation functions, such as $\tanh(x)$, $\text{sigmoid}(x)$, $\text{softplus}(x)$, etc., are all multiplicity one, the theoretical analysis sheds light on practical training.

## 6 DISCUSSION

In this work, we have shown that the characteristic of the activation function, i.e., multiplicity, is a key factor to understanding the complexity of NN output and the weight condensation at initial training. The condensation restricts the NN to be effectively low-capacity at the initial training stage, even for finite-width NNs. During the training, the NN increases its capacity to better fit the data, leading to a potential explanation for their good generalization in practical problems. This work also serves as a starting point for further studying the condensation for multiple-layer neural networks throughout the training process.

As the scale of parameter initialization becomes larger, the condensation becomes weaker. The understanding from a complete condensation would benefit our understanding of the training process in common initialization, that is there is an effect, although not so strong, that can limit the complexity of the neural network in the initial training.

How small the initialization should be in order to see a clear condensation is studied in Luo et al. (2021) for two-layer ReLU NNs with infinite width. For general activation functions, the regime of the initialization for condensation depends on the NN width. A further study of the phase diagram for finite width NNs would be important.

For general multiplicity with high-dimensional input data, the theoretical analysis for the initial condensation is a very difficult problem, which is equivalent to count the number of the roots of a high-order high-dimensional polynomial with a special structure originated from NNs.

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

# A  APPENDIX

## A.1  BASIC DEFINITIONS

In this study, we first consider the neural network with 2 hidden layers,

A two-layer NN is

$$f_{\boldsymbol{\theta}}(\boldsymbol{x}) = \sum_{j=1}^{m} a_j \sigma(\boldsymbol{w}_j \cdot \boldsymbol{x}), \tag{6}$$

where $\sigma(\cdot)$ is the activation function, $\boldsymbol{w}_j = (\bar{\boldsymbol{w}}_j, \boldsymbol{b}_j) \in \mathbb{R}^{d+1}$ is the neuron feature including the input weight and bias terms, and $\boldsymbol{x} = (\bar{\boldsymbol{x}}, 1) \in \mathbb{R}^{d+1}$ is combination of the input sample and scalar 1, $\boldsymbol{\theta}$ is the set of all parameters, i.e., $\{a_j, \boldsymbol{w}_j\}_{j=1}^{m}$. For simplicity, **we call $\boldsymbol{w}_j$ as input weight or weight** and $\boldsymbol{x}$ as input sample.

Then, we consider the neural network with $l$ hidden layers,

$$\boldsymbol{x}^{[0]} = (\boldsymbol{x}, 1), \quad \boldsymbol{x}^{[1]} = (\sigma(\boldsymbol{W}^{[1]}\boldsymbol{x}^{[0]}), 1), \quad \boldsymbol{x}^{[l]} = (\sigma(\boldsymbol{W}^{[l]}\boldsymbol{x}^{[l-1]}), 1), \text{ for } l \in \{2, 3, ..., L\}$$
$$f(\boldsymbol{\theta}, \boldsymbol{x}) = \frac{1}{\alpha}\mathbf{a}^{\mathsf{T}}\boldsymbol{x}^{[L]} \triangleq f_{\boldsymbol{\theta}}(\boldsymbol{x}), \tag{7}$$

where $\boldsymbol{W}^{[l]} = (\bar{\boldsymbol{W}}^{[l]}, \boldsymbol{b}^{[l]}) \in \mathbb{R}^{(m_l \times m_{l-1})}$, and $m_l$ represents the dimension of the $l$-th hidden layer. The initialization of $\boldsymbol{W}_{k,k'}^{[l]}, l \in \{1, 2, 3, ..., L\}$ and $\mathbf{a}_k$ obey normal distribution $\boldsymbol{W}_{k,k'}^{[l]} \sim \mathcal{N}(0, \beta_l^2)$ for $l \in \{1, 2, 3, ..., L\}$ and $\mathbf{a}_k \sim \mathcal{N}(0, \beta_{L+1}^2)$.

The loss function is mean squared error given below,

$$R_s(\boldsymbol{\theta}) = \frac{1}{2n} \sum_{i=1}^{n} (f_{\boldsymbol{\theta}}(\boldsymbol{x}_i) - y_i)^2. \tag{8}$$

For simplification, we denote $f_{\boldsymbol{\theta}}(\boldsymbol{x})$ as $f$ in following.

## A.2  DERIVATIONS FOR CONCERNED QUANTITIES

### A.2.1  NEURAL NETWORKS WITH THREE HIDDEN LAYERS

In order to better understand the gradient of the parameter matrix of the multi-layer neural network, we first consider the case of the three-layer neural network,

$$f_{\boldsymbol{\theta}}(\boldsymbol{x}) := \boldsymbol{a}^{\mathsf{T}}\sigma(\boldsymbol{W}^{[2]}\sigma(\boldsymbol{W}^{[1]}\boldsymbol{x})), \tag{9}$$

with the mean squared error as the loss function,

$$R_s(\boldsymbol{\theta}) = \frac{1}{2n} \sum_{i=1}^{n} (f_{\boldsymbol{\theta}}(\boldsymbol{x}_i) - y_i)^2. \tag{10}$$

We calculate $\frac{\mathrm{d}f}{\mathrm{d}\boldsymbol{W}^{[2]}}$ and $\frac{\mathrm{d}f}{\mathrm{d}\boldsymbol{W}^{[1]}}$ respectively, using differential form,

$$\mathrm{d}f = \mathrm{tr}((\frac{\partial f}{\partial \boldsymbol{x}})^{\mathsf{T}}\mathrm{d}f). \tag{11}$$

We consider $\frac{\mathrm{d}f}{\mathrm{d}\boldsymbol{W}^{[2]}}$ first,

$$\begin{aligned}
\mathrm{d}f &= \mathrm{tr}\{\mathrm{d}(\boldsymbol{a}^{\mathsf{T}}\sigma(\boldsymbol{W}^{[2]}\boldsymbol{x}^{[1]}))\} \\
&= \mathrm{tr}\{\boldsymbol{a}^{\mathsf{T}}\mathrm{d}(\sigma(\boldsymbol{W}^{[2]}\boldsymbol{x}^{[1]}))\} \\
&= \mathrm{tr}\{\boldsymbol{a}^{\mathsf{T}}\sigma^{'}(\boldsymbol{W}^{[2]}\boldsymbol{x}^{[1]}) \odot \mathrm{d}\boldsymbol{W}^{[2]}\boldsymbol{x}^{[1]}\} \\
&= \mathrm{tr}\{(\boldsymbol{a} \odot \sigma^{'}(\boldsymbol{W}^{[2]}\boldsymbol{x}^{[1]})^{\mathsf{T}}\mathrm{d}\boldsymbol{W}^{[2]}\boldsymbol{x}^{[1]}\} \\
&= \mathrm{tr}\{\boldsymbol{x}^{[1]}(\boldsymbol{a} \odot \sigma^{'}(\boldsymbol{W}^{[2]}\boldsymbol{x}^{[1]})^{\mathsf{T}}\mathrm{d}\boldsymbol{W}^{[2]}\} \\
&= \mathrm{tr}\{((\boldsymbol{a} \odot \sigma^{'}(\boldsymbol{W}^{[2]}\boldsymbol{x}^{[1]}))\boldsymbol{x}^{[1]\mathsf{T}})^{\mathsf{T}}\mathrm{d}\boldsymbol{W}^{[2]}\},
\end{aligned} \tag{12}$$

where $\odot$ is Hadamard Product, and it is the multiplication of matrix elements of the same position. Hence,

$$\begin{aligned}
\frac{\mathrm{d}f}{\mathrm{d}\boldsymbol{W}^{[2]}} &= (\boldsymbol{a} \odot \sigma^{'}(\boldsymbol{W}^{[2]}\boldsymbol{x}^{[1]}))\boldsymbol{x}^{[1]\mathsf{T}} \\
&= \mathrm{diag}\{\sigma^{'}(\boldsymbol{W}^{[2]}\boldsymbol{x}^{[1]})\}\boldsymbol{a}\boldsymbol{x}^{[1]\mathsf{T}}.
\end{aligned} \tag{13}$$

Then, we consider $\frac{\mathrm{d}f}{\mathrm{d}\boldsymbol{W}^{[1]}}$,

$$\begin{aligned}
\mathrm{d}f &= \mathrm{tr}\{(\boldsymbol{a} \odot \sigma^{'}(\boldsymbol{W}^{[2]}\boldsymbol{x}^{[1]}))^{\mathsf{T}}\boldsymbol{W}^{[2]}\mathrm{d}\sigma(\boldsymbol{W}^{[1]}\boldsymbol{x})\} \\
&= \mathrm{tr}\{(\boldsymbol{W}^{[2]\mathsf{T}}(\boldsymbol{a} \odot \sigma^{'}(\boldsymbol{W}^{[2]}\boldsymbol{x}^{[1]})))^{\mathsf{T}}\sigma^{'}(\boldsymbol{W}^{[1]}\boldsymbol{x}) \odot \mathrm{d}(\boldsymbol{W}^{[1]}\boldsymbol{x})\} \\
&= \mathrm{tr}\{((\boldsymbol{W}^{[2]\mathsf{T}}(\boldsymbol{a} \odot \sigma^{'}(\boldsymbol{W}^{[2]}\boldsymbol{x}^{[1]})) \odot \sigma^{'}(\boldsymbol{W}^{[1]}\boldsymbol{x}))^{\mathsf{T}}\mathrm{d}(\boldsymbol{W}^{[1]}\boldsymbol{x})\} \\
&= \mathrm{tr}\{[((\boldsymbol{W}^{[2]\mathsf{T}}(\boldsymbol{a} \odot \sigma^{'}(\boldsymbol{W}^{[2]}\boldsymbol{x}^{[1]})) \odot \sigma^{'}(\boldsymbol{W}^{[1]}\boldsymbol{x}))\boldsymbol{x}^{\mathsf{T}}]^{\mathsf{T}}\mathrm{d}(\boldsymbol{W}^{[1]})\}.
\end{aligned} \tag{14}$$

Hence, we have,

$$\begin{aligned}
\frac{\mathrm{d}f}{\mathrm{d}\boldsymbol{W}^{[1]}} &= ((\boldsymbol{W}^{[2]\mathsf{T}}(\boldsymbol{a} \odot \sigma^{'}(\boldsymbol{W}^{[2]}\boldsymbol{x}^{[1]}))) \odot \sigma^{'}(\boldsymbol{W}^{[1]}\boldsymbol{x}))\boldsymbol{x}^{\mathsf{T}} \\
&= \mathrm{diag}\{\sigma^{'}(\boldsymbol{W}^{[1]}\boldsymbol{x})\}\boldsymbol{W}^{[2]\mathsf{T}}\mathrm{diag}\{\sigma^{'}(\boldsymbol{W}^{[2]}\boldsymbol{x}^{[1]})\}\boldsymbol{a}\boldsymbol{x}^{\mathsf{T}}.
\end{aligned} \tag{15}$$

Through the chain rule, we can get the evolution equation of $\boldsymbol{W}^{[1]}$ and $\boldsymbol{W}^{[2]}$,

$$\begin{aligned}
\frac{\mathrm{d}\boldsymbol{W}^{[1]}}{\mathrm{d}t} &= -\frac{\mathrm{d}R_s(\boldsymbol{\theta})}{\mathrm{d}\boldsymbol{W}^{[1]}} \\
&= -\frac{1}{n}\sum_{i=1}^{n}(f(\boldsymbol{\theta}, \boldsymbol{x}_i) - y_i)\frac{\mathrm{d}f}{\mathrm{d}\boldsymbol{W}^{[1]}} \\
&= -\frac{1}{n}\sum_{i=1}^{n}(f(\boldsymbol{\theta}, \boldsymbol{x}_i) - y_i)\mathrm{diag}\{\sigma^{'}(\boldsymbol{W}^{[1]}\boldsymbol{x}_i)\}\boldsymbol{W}^{[2]\mathsf{T}}\mathrm{diag}\{\sigma^{'}(\boldsymbol{W}^{[2]}\boldsymbol{x}_i^{[1]})\}\boldsymbol{a}\boldsymbol{x}_i^{\mathsf{T}},
\end{aligned} \tag{16}$$

and

$$\begin{aligned}
\frac{\mathrm{d}\boldsymbol{W}^{[2]}}{\mathrm{d}t} &= -\frac{\mathrm{d}R_s(\boldsymbol{\theta})}{\mathrm{d}\boldsymbol{W}^{[2]}} \\
&= -\frac{1}{n}\sum_{i=1}^{n}(f(\boldsymbol{\theta}, \boldsymbol{x}_i) - y_i)\frac{\mathrm{d}f}{\mathrm{d}\boldsymbol{W}^{[1]}} \\
&= -\frac{1}{n}\sum_{i=1}^{n}(f(\boldsymbol{\theta}, \boldsymbol{x}_i) - y_i)\mathrm{diag}\{\sigma^{'}(\boldsymbol{W}^{[2]}\boldsymbol{x}_i^{[1]})\}\boldsymbol{a}\boldsymbol{x}_i^{[1]\mathsf{T}}.
\end{aligned} \tag{17}$$

### A.2.2 $L$ HIDDEN LAYERS CONDITION

And, we consider the neural network with $L$ hidden layers,

$$
\begin{aligned}
\mathrm{d}f &= \mathrm{tr}\{\mathrm{d}\boldsymbol{a}^\mathsf{T}\mathrm{d}\sigma(\boldsymbol{W}^{[L]}\boldsymbol{x}^{[L-1]})\} \\
&= \mathrm{tr}\{(\boldsymbol{a}\odot\sigma^{'}(\boldsymbol{W}^{[L]}\boldsymbol{x}^{[L-1]}))^\mathsf{T}\mathrm{d}\boldsymbol{W}^{[L]}\sigma(\boldsymbol{W}^{[L-1]}\boldsymbol{x}^{[L-2]})\} \\
&= \mathrm{tr}\{(\boldsymbol{W}^{[L]\mathsf{T}}\Lambda_L)^\mathsf{T}\sigma^{'}(\boldsymbol{W}^{[L-1]}\boldsymbol{x}^{[L-2]})\odot\mathrm{d}\boldsymbol{W}^{[L-1]}\sigma(\boldsymbol{W}^{[L-2]}\boldsymbol{x}^{[L-3]})\} \\
&= \mathrm{tr}\{((\boldsymbol{W}^{[L]\mathsf{T}}\Lambda_L)\odot\sigma^{'}(\boldsymbol{W}^{[L-1]}\boldsymbol{x}^{[L-2]}))^\mathsf{T}\boldsymbol{W}^{[L-1]}\mathrm{d}\sigma(\boldsymbol{W}^{[L-2]}\boldsymbol{x}^{[L-3]})\} \\
&= (\boldsymbol{W}^{[L-1]\mathsf{T}}\Lambda_{L-1})^\mathsf{T}\mathrm{d}\sigma(\boldsymbol{W}^{[L-2]}\boldsymbol{x}^{[L-3]}) \\
&= \ldots \\
&= \mathrm{tr}\{\Lambda_k^\mathsf{T}\mathrm{d}\boldsymbol{W}^{[k]}\boldsymbol{x}^{[k-1]}\} \\
&= \mathrm{tr}\{(\Lambda_k\boldsymbol{x}^{[k-1]\mathsf{T}})^\mathsf{T}\mathrm{d}\boldsymbol{W}^{[k]}\},
\end{aligned}
\tag{18}
$$

where $\Lambda_l \triangleq (\boldsymbol{W}^{[l+1]\mathsf{T}}\Lambda_{l+1})\odot\sigma^{'}(\boldsymbol{W}^{[l]}\boldsymbol{x}^{[l-1]})$ for $l = k, k+1\ldots L-1$ and $\Lambda_L \triangleq \boldsymbol{a}\odot\sigma^{'}(\boldsymbol{W}^{[L]}\boldsymbol{x}^{[L-1]})$.

Hence, we get,

$$
\frac{\mathrm{d}f}{\mathrm{d}\boldsymbol{W}^{[k]}} = \Lambda_k\boldsymbol{x}^{[k-1]\mathsf{T}}.
\tag{19}
$$

Through the chain rule, we can get the evolution equation of $\boldsymbol{W}^{[k]}$,

$$
\begin{aligned}
\frac{\mathrm{d}\boldsymbol{W}^{[k]}}{\mathrm{d}t} &= -\frac{\mathrm{d}R_s(\boldsymbol{\theta})}{\mathrm{d}\boldsymbol{W}^{[k]}} \\
&= -\frac{1}{n}\sum_{i=1}^{n}(f(\boldsymbol{\theta},\boldsymbol{x}_i)-y_i)\frac{\mathrm{d}f}{\mathrm{d}\boldsymbol{W}^{[k]}} \\
&= -\frac{1}{n}\sum_{i=1}^{n}(f(\boldsymbol{\theta},\boldsymbol{x}_i)-y_i)\Lambda_k\boldsymbol{x}_i^{[k-1]\mathsf{T}}.
\end{aligned}
\tag{20}
$$

Through $\boldsymbol{a}\odot\sigma'(\boldsymbol{W}\boldsymbol{x}) = \mathrm{diag}\{\sigma^{'}(\boldsymbol{W}\boldsymbol{x})\}\boldsymbol{a}$,

Finally, the dynamic system can be obtained:

$$
\dot{\boldsymbol{a}} = \frac{\mathrm{d}\mathbf{a}}{\mathrm{d}t} = -\frac{1}{n}\sum_{i=1}^{n}\boldsymbol{x}_i^{[L]}\left(f(\boldsymbol{\theta},\boldsymbol{x}_i)-y_i\right),
$$

$$
\dot{\boldsymbol{W}}^{[L]} = \frac{\mathrm{d}\mathbf{W}^{[L]}}{\mathrm{d}t} = -\frac{1}{n}\sum_{i=1}^{n}\mathrm{diag}\{\sigma'(\boldsymbol{W}^{[L]}\boldsymbol{x}_i^{[L-1]})\}\mathbf{a}\boldsymbol{x}_i^{[L-1]\mathsf{T}}\left(f(\boldsymbol{\theta},\boldsymbol{x}_i)-y_i\right),
$$

$$
\dot{\boldsymbol{W}}^{[k]} = \frac{\mathrm{d}\mathbf{W}^{[k]}}{\mathrm{d}t} = -\frac{1}{n}\sum_{i=1}^{n}\mathrm{diag}\{\sigma'(\boldsymbol{W}^{[k]}\boldsymbol{x}_i^{[k-1]})\}E^{[k+1:L]}\mathbf{a}\boldsymbol{x}_i^{[k-1]\mathsf{T}}\left(f(\boldsymbol{\theta},\boldsymbol{x}_i)-y_i\right)\ \forall i\in[1:L-1],
$$
$$
\tag{21}
$$

where we use $E^l(\boldsymbol{x}) = \boldsymbol{W}^{[l]\mathsf{T}}\mathrm{diag}\{\sigma'(\boldsymbol{W}^{[l]}\boldsymbol{x}^{[l-1]})\}$. And $E^{[q:p]} = E^q E^{q+1}\ldots E^p$.

Let $r_{k,j} = \|\boldsymbol{W}_j^{[k]}\|_2$. We have

$$
\frac{\mathrm{d}}{\mathrm{d}t}|r_{k,j}|^2 = \frac{\mathrm{d}}{\mathrm{d}t}\|\boldsymbol{W}_j^{[k]}\|^2.
\tag{22}
$$

Then we obtain

$$
\dot{r}_{k,j}r_{k,j} = \dot{\boldsymbol{W}}_j^{[k]}\cdot\boldsymbol{W}_j^{[k]}.
\tag{23}
$$

Finally, we get

$$
\begin{aligned}
\dot{r}_{k,j} = \frac{\mathrm{d}r_{k,j}}{\mathrm{d}t} &= \dot{\boldsymbol{W}}_j^{[k]}\cdot\boldsymbol{W}_j^{[k]}/r_{k,j} \\
&= \dot{\boldsymbol{W}}_j^{[k]}\cdot\boldsymbol{u}_{k,j},
\end{aligned}
\tag{24}
$$

where $\boldsymbol{u}_{k,j} = \frac{\boldsymbol{W}_j^{[k]}}{r_{k,j}}$ is a unit vector. Then we have,

$$
\begin{aligned}
\dot{\boldsymbol{u}}_{k,j} = \frac{\mathrm{d}\boldsymbol{u}_{k,j}}{\mathrm{d}t} &= \frac{\mathrm{d}}{\mathrm{d}t}\left(\frac{\boldsymbol{W}_j^{[k]}}{r_{k,j}}\right) \\
&= \frac{\dot{\boldsymbol{W}}_j^{[k]} r_{k,j} - \boldsymbol{W}_j^{[k]} \dot{r}_{k,j}}{r_{k,j}^2} \\
&= \frac{\dot{\boldsymbol{W}}_j^{[k]} r_{k,j} - \boldsymbol{W}_j^{[k]}(\dot{\boldsymbol{W}}_j^{[k]} \cdot \boldsymbol{u}_{k,j})}{r_{k,j}^2} \\
&= \frac{\dot{\boldsymbol{W}}_j^{[k]} - \boldsymbol{u}_{k,j}(\dot{\boldsymbol{W}}_j^{[k]} \cdot \boldsymbol{u}_{k,j})}{r_{k,j}}.
\end{aligned}
\tag{25}
$$

To conclude, the quantities we concern are summarized as follows,

$$
\begin{cases}
\dot{\boldsymbol{a}} = -\frac{1}{n}\sum_{i=1}^{n} \boldsymbol{x}_i^{[L]}\left(f(\boldsymbol{\theta},\boldsymbol{x}_i) - y_i\right) & (26) \\[2ex]
\dot{\boldsymbol{W}}^{[L]} = -\frac{1}{n}\sum_{i=1}^{n} \mathrm{diag}\{\sigma'(\boldsymbol{W}^{[L]}\boldsymbol{x}_i^{[L-1]})\}\mathbf{a}\boldsymbol{x}_i^{[L-1]\mathsf{T}}\left(f(\boldsymbol{\theta},\boldsymbol{x}_i) - y_i\right), & (27) \\[2ex]
\dot{\boldsymbol{W}}^{[k]} = -\frac{1}{n}\sum_{i=1}^{n} \mathrm{diag}\{\sigma'(\boldsymbol{W}^{[k]}\boldsymbol{x}_i^{[k-1]})\}E^{[k+1:L]}\mathbf{a}\boldsymbol{x}_i^{[k-1]\mathsf{T}}\left(f(\boldsymbol{\theta},\boldsymbol{x}_i) - y_i\right) \ \forall i \in [1:L-1] & (28) \\[2ex]
\dot{r}_{k,j} = \dot{\boldsymbol{W}}_j^{[k]} \cdot \boldsymbol{u}_{k,j} & (29) \\[2ex]
\dot{\boldsymbol{u}}_{k,j} = \frac{\dot{\boldsymbol{W}}_j^{[k]} - \boldsymbol{u}_{k,j}(\dot{\boldsymbol{W}}_j^{[k]} \cdot \boldsymbol{u}_{k,j})}{r_{k,j}}, & (30)
\end{cases}
$$

where we use $E^l(\boldsymbol{x}) = \boldsymbol{W}^{[l]\mathsf{T}}\mathrm{diag}\{\sigma'(\boldsymbol{W}^{[l]}\boldsymbol{x}^{[l-1]})\}$. And $E^{[q:p]} = E^q E^{q+1}...E^p$.

### A.2.3 PROVE FOR $\mathcal{P}w$ IN 5.2

We calculate $\mathcal{P}\boldsymbol{w} \overset{\text{leading order}}{\approx} \mathcal{Q}\boldsymbol{w}$ as following,

$$
\begin{aligned}
\mathcal{P}\boldsymbol{w} \approx \mathcal{Q}\boldsymbol{w} :={}& -\frac{1}{n}\sum_{i=1}^{n} e_i \boldsymbol{x}_i^{[k-1]}[\mathrm{diag}\{\sigma'(\boldsymbol{W}^{[k]}\boldsymbol{x}_i^{[k-1]})\}(E^{[k+1:L]}\mathbf{a})]_j \\
& + (\frac{1}{n}\sum_{i=1}^{n} e_i \boldsymbol{x}_i^{[k-1]}[\mathrm{diag}\{\sigma'(\boldsymbol{W}^{[k]}\boldsymbol{x}_i^{[k-1]})\}(E^{[k+1:L]}\mathbf{a})]_j \cdot \boldsymbol{u})\boldsymbol{u} \\
={}& -\frac{1}{n}\sum_{i=1}^{n} e_i \boldsymbol{x}_i^{[k-1]}[\mathrm{diag}\{\frac{\sigma^{(p)}(\mathbf{0})}{(p-1)!} \odot (\boldsymbol{W}^{[k]}\boldsymbol{x}_i^{[k-1]})^{p-1}\}(E^{[k+1:L]}\mathbf{a})]_j \\
& + (\frac{1}{n}\sum_{i=1}^{n} e_i \boldsymbol{x}_i^{[k-1]}[\mathrm{diag}\{\frac{\sigma^{(p)}(\mathbf{0})}{(p-1)!} \odot (\boldsymbol{W}^{[k]}\boldsymbol{x}_i^{[k-1]})^{p-1}\}(E^{[k+1:L]}\mathbf{a})]_j \cdot \boldsymbol{u})\boldsymbol{u} \\
={}& -\frac{1}{n}\sum_{i=1}^{n} e_i \boldsymbol{x}_i^{[k-1]}[\mathrm{diag}\{(\boldsymbol{W}^{[k]}\boldsymbol{x}_i^{[k-1]})^{p-1}\}\,\mathrm{diag}\{\frac{\sigma^{(p)}(\mathbf{0})}{(p-1)!}\}(E^{[k+1:L]}\mathbf{a})]_j \\
& + (\frac{1}{n}\sum_{i=1}^{n} e_i \boldsymbol{x}_i^{[k-1]}[\mathrm{diag}\{(\boldsymbol{W}^{[k]}\boldsymbol{x}_i^{[k-1]})^{p-1}\}\,\mathrm{diag}\{\frac{\sigma^{(p)}(\mathbf{0})}{(p-1)!}\}(E^{[k+1:L]}\mathbf{a})]_j \cdot \boldsymbol{u})\boldsymbol{u} \\
={}& -\frac{1}{n}\sum_{i=1}^{n} e_i \boldsymbol{x}_i^{[k-1]}[\mathrm{diag}\{(\boldsymbol{W}^{[k]}\boldsymbol{x}_i^{[k-1]})^{p-1}\}]_j\,\mathrm{diag}\{\frac{\sigma^{(p)}(\mathbf{0})}{(p-1)!}\}(E^{[k+1:L]}\mathbf{a}) \\
& + (\frac{1}{n}\sum_{i=1}^{n} e_i \boldsymbol{x}_i^{[k-1]}[\mathrm{diag}\{(\boldsymbol{W}^{[k]}\boldsymbol{x}_i^{[k-1]})^{p-1}\}]_j\,\mathrm{diag}\{\frac{\sigma^{(p)}(\mathbf{0})}{(p-1)!}\}(E^{[k+1:L]}\mathbf{a}) \cdot \boldsymbol{u})\boldsymbol{u} \\
={}& -(\frac{1}{n}\sum_{i=1}^{n} e_i \boldsymbol{x}_i^{[k-1]}(\boldsymbol{W}_j^{[k]}\boldsymbol{x}_i^{[k-1]})^{p-1})[\mathrm{diag}\{\frac{\sigma^{(p)}(\mathbf{0})}{(p-1)!}\}(E^{[k+1:L]}\mathbf{a})]_j \\
& + ((\frac{1}{n}\sum_{i=1}^{n} e_i \boldsymbol{x}_i^{[k-1]}(\boldsymbol{W}_j^{[k]}\boldsymbol{x}_i^{[k-1]})^{p-1})[\mathrm{diag}\{\frac{\sigma^{(p)}(\mathbf{0})}{(p-1)!}\}(E^{[k+1:L]}\mathbf{a})]_j \cdot \boldsymbol{u})\boldsymbol{u},
\end{aligned}
\tag{31}
$$

where $(\cdot)^{p-1}$ and $\sigma^p(\cdot)$ operate on component here.

### A.3 The Verification of the initial stage

We put the loss of the experiments in the main text here to show that they are indeed in the initial stage of training by the definition.

As is shown in Fig.7 and Fig.8, at the number of steps I drew the graphs in the article, loss satisfies the definition of the initial stage, so we consider that they are in the initial stage of training.

Only the epoch number could hardly reflect the initial stage, which also depends on the learning rate. Therefore, we use the loss to indicate the initial stage. Learning rate is not a sensitive to the appearance of condensation. However, a small learning rate enables us to clearly observe the condensation process in the initial stage under a gradient flow training. For example, when the learning rate is relatively small, the initial stage of training may be relatively long, while when the learning rate is relatively large, the initial stage of training may be relatively small.

We empirically find that to ensure the training process follows a gradient follow, where the loss decays monotonically, we have to select a smaller learning rate for large multiplicity $p$. Therefore, it looks like we have a longer training in our experiments with large $p$. Note that for a small learning rate in the experiments of small $p$, we can observe similar phenomena.

In all subsequent experiments in the Appendix, we will no longer show the loss graph of each experiment one by one, but we make sure that they are indeed in the initial stage of training.

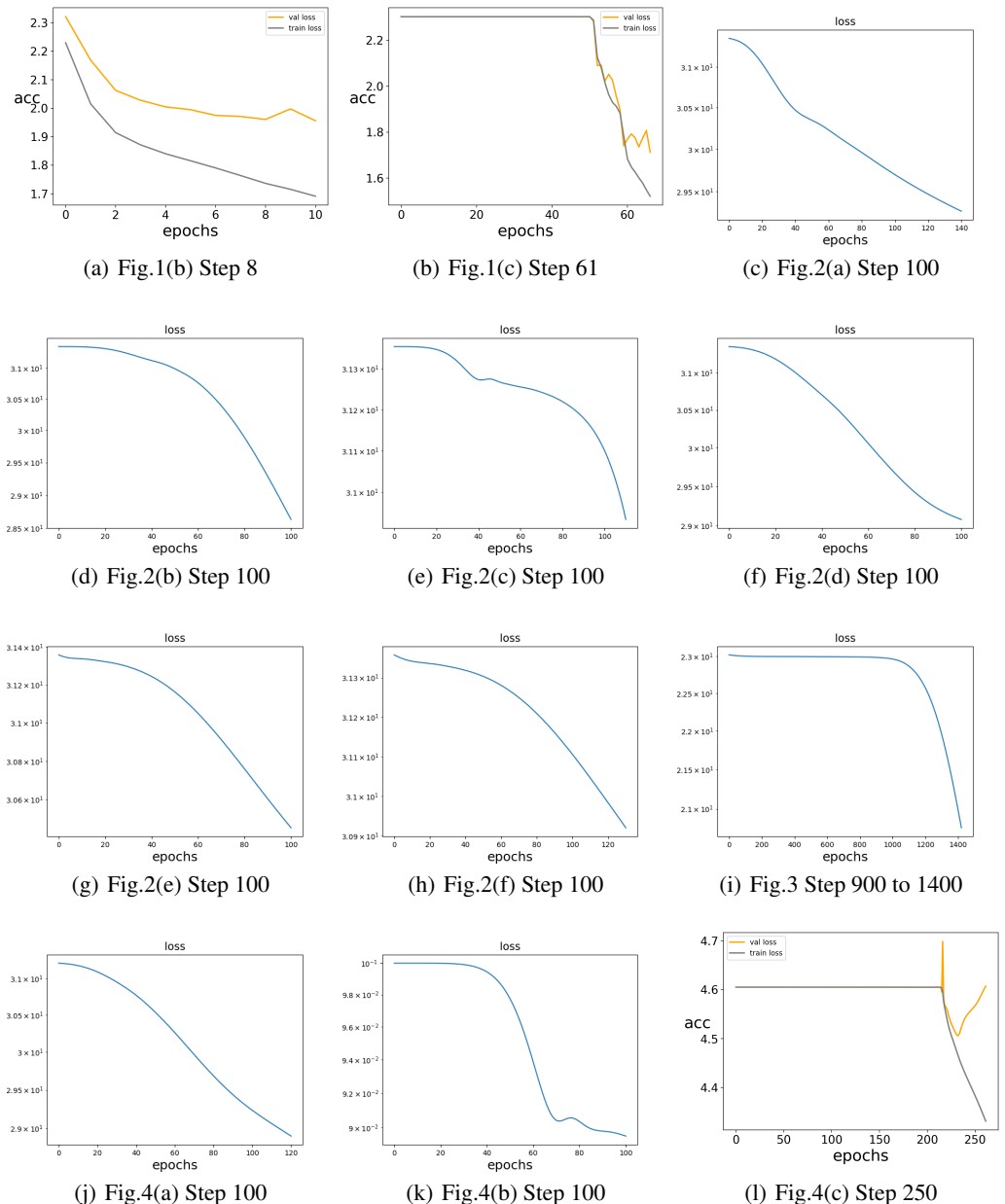

Figure 7: Losses from Fig. 1 to Fig.4. The original pictures and the numbers of steps corresponding to each sub-picture are written in the sub-captions.

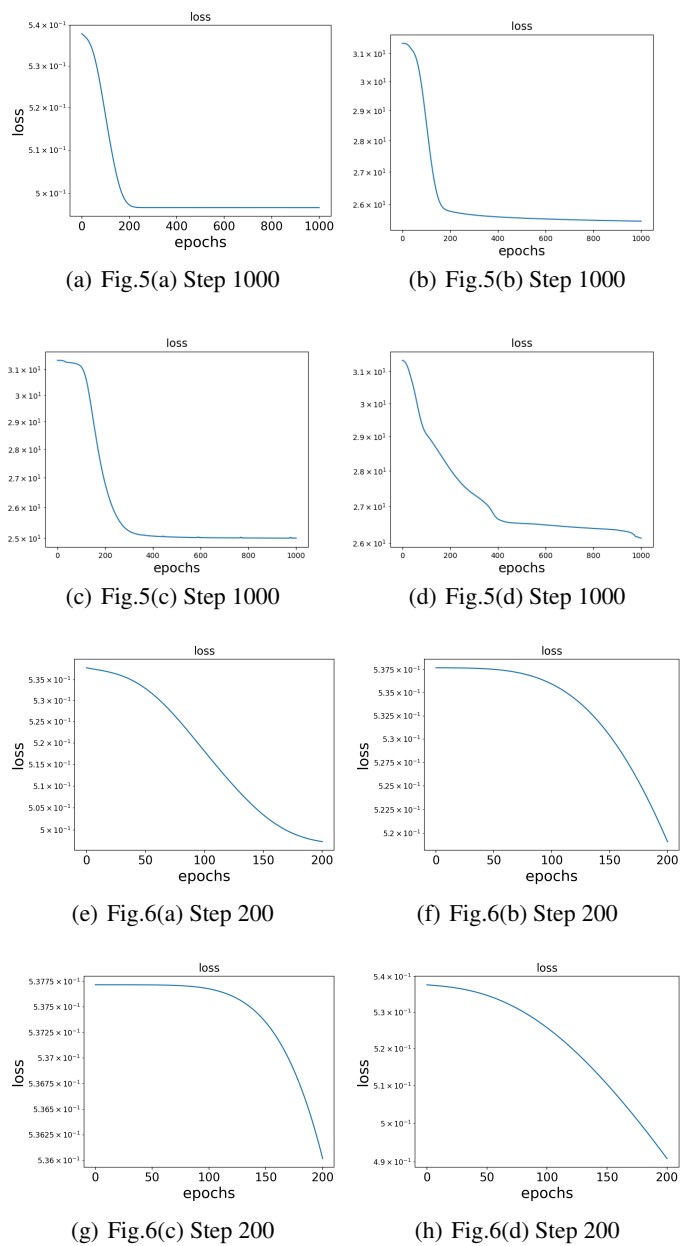

Figure 8: Losses from Fig. 5 to Fig.6. The original pictures and the numbers of steps corresponding to each sub-picture are written in the sub-captions.

## A.4 Several steps during the evolution of condensation at the initial stage

In the article, we only give the results of the last step of each condense, while the details of the evolution of condensation are lacking, which may provide us a better understanding. Therefore, we show these details in Fig. 9, Fig. 10, Fig. 11 and Fig. 12, which also further illustrate the rationality of the experimental results and facilitate the understanding of the evolution of condensation in the initial stage.

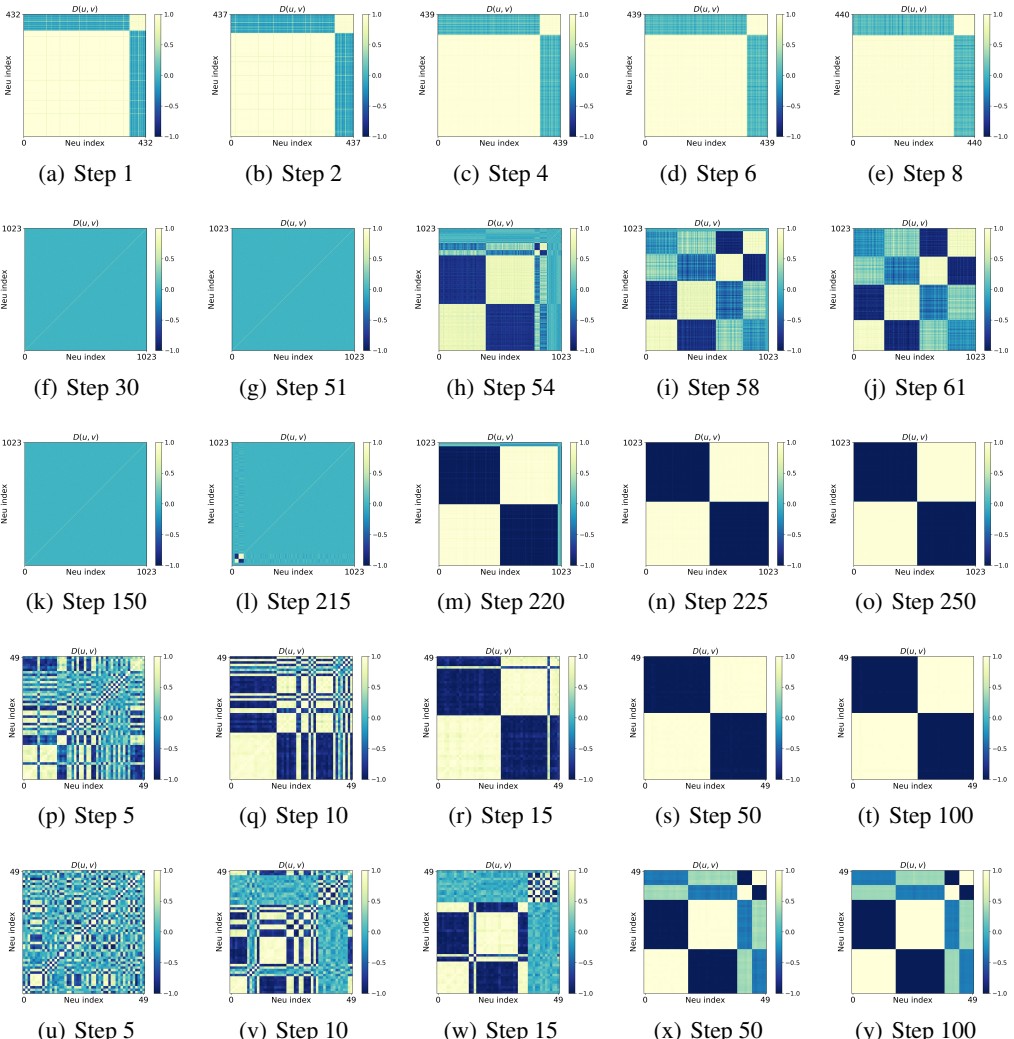

Figure 9: Evolution of condensation from Fig. 1(b) to Fig. 2(b) and Fig. 1(c). The evolution from the first row to the fifth row are corresponding to the Fig. 1(b), Fig. 1(c), Fig. 4(c), Fig. 2(a), Fig. 2(b). The numbers of evolutionary steps are shown in the sub-captions, where sub-figures in the last row are the epochs in the article.

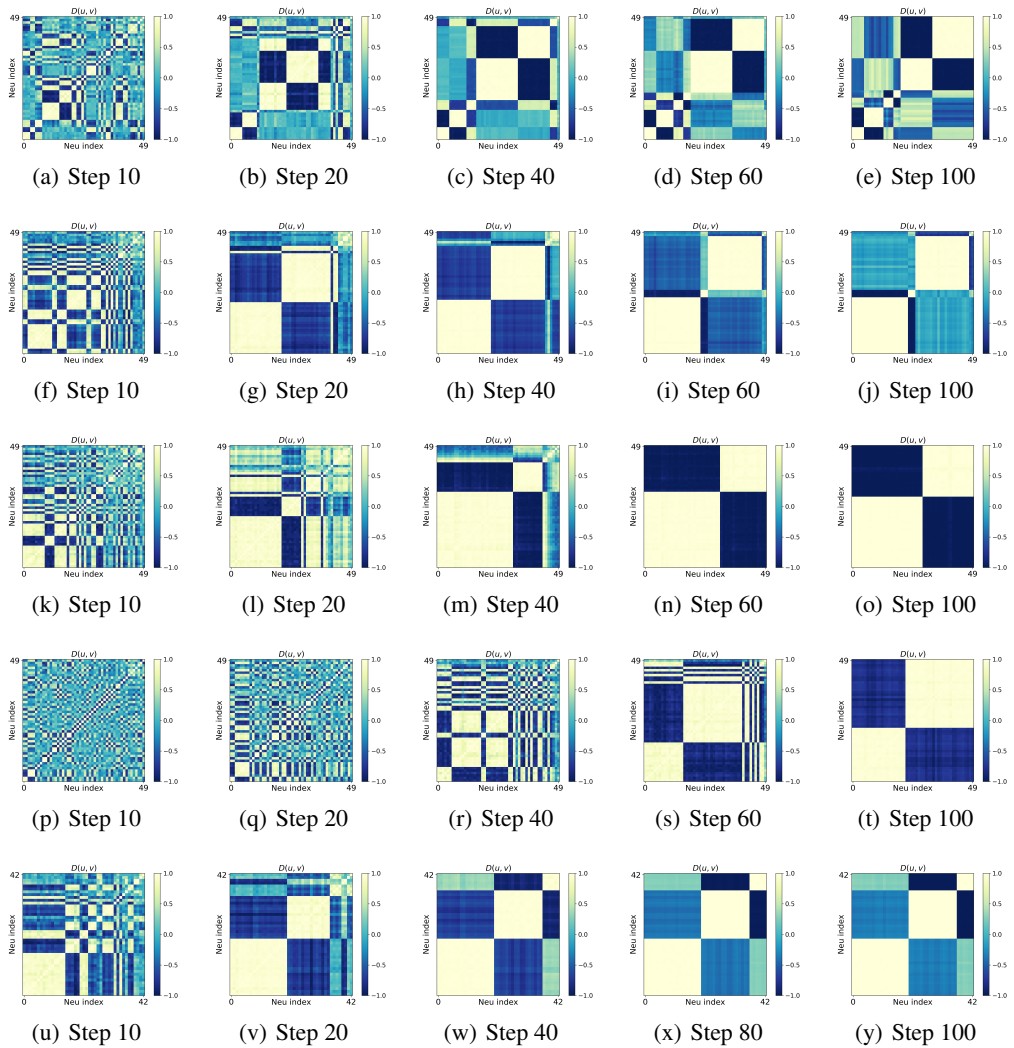

(a) Step 10    (b) Step 20    (c) Step 40    (d) Step 60    (e) Step 100

(f) Step 10    (g) Step 20    (h) Step 40    (i) Step 60    (j) Step 100

(k) Step 10    (l) Step 20    (m) Step 40    (n) Step 60    (o) Step 100

(p) Step 10    (q) Step 20    (r) Step 40    (s) Step 60    (t) Step 100

(u) Step 10    (v) Step 20    (w) Step 40    (x) Step 80    (y) Step 100

Figure 10: Evolution of condensation from Fig. 2(c) to 2(f) and Fig. 4(a). The evolution from the first row to the fifth row are corresponding to the Fig. 2(c), Fig. 2(d), Fig. 2(e), Fig. 2(f), Fig. 4(a). The numbers of evolutionary steps are shown in the sub-captions, where sub-figures in the last row are the epochs in the article.

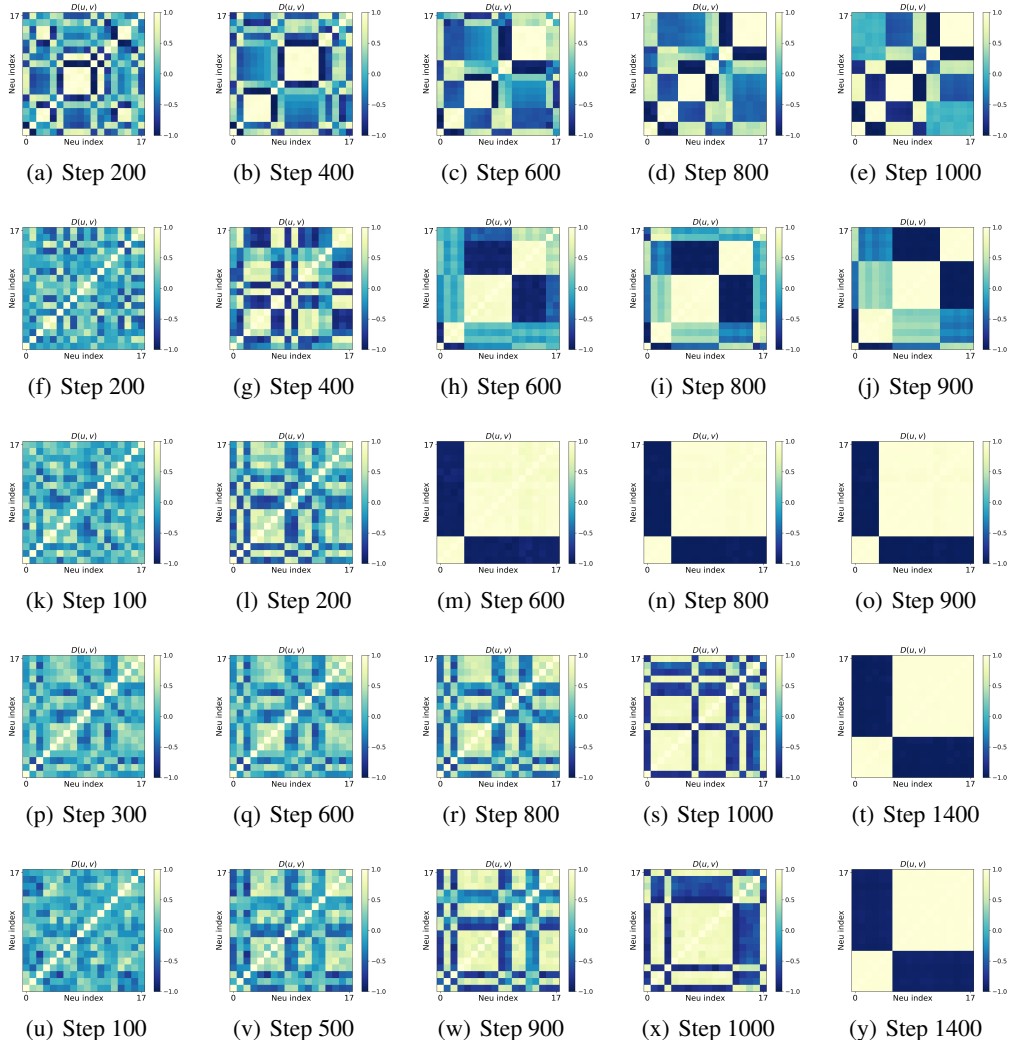

Figure 11: Evolution of condensation from Fig. 3(a) to 3(e). The evolution from the first row to the fifth row are corresponding to the Fig. 3(a), Fig. 3(b), Fig. 3(c), Fig. 3(d), Fig. 3(e). The numbers of evolutionary steps are shown in the sub-captions, where sub-figures in the last row are the epochs in the article.

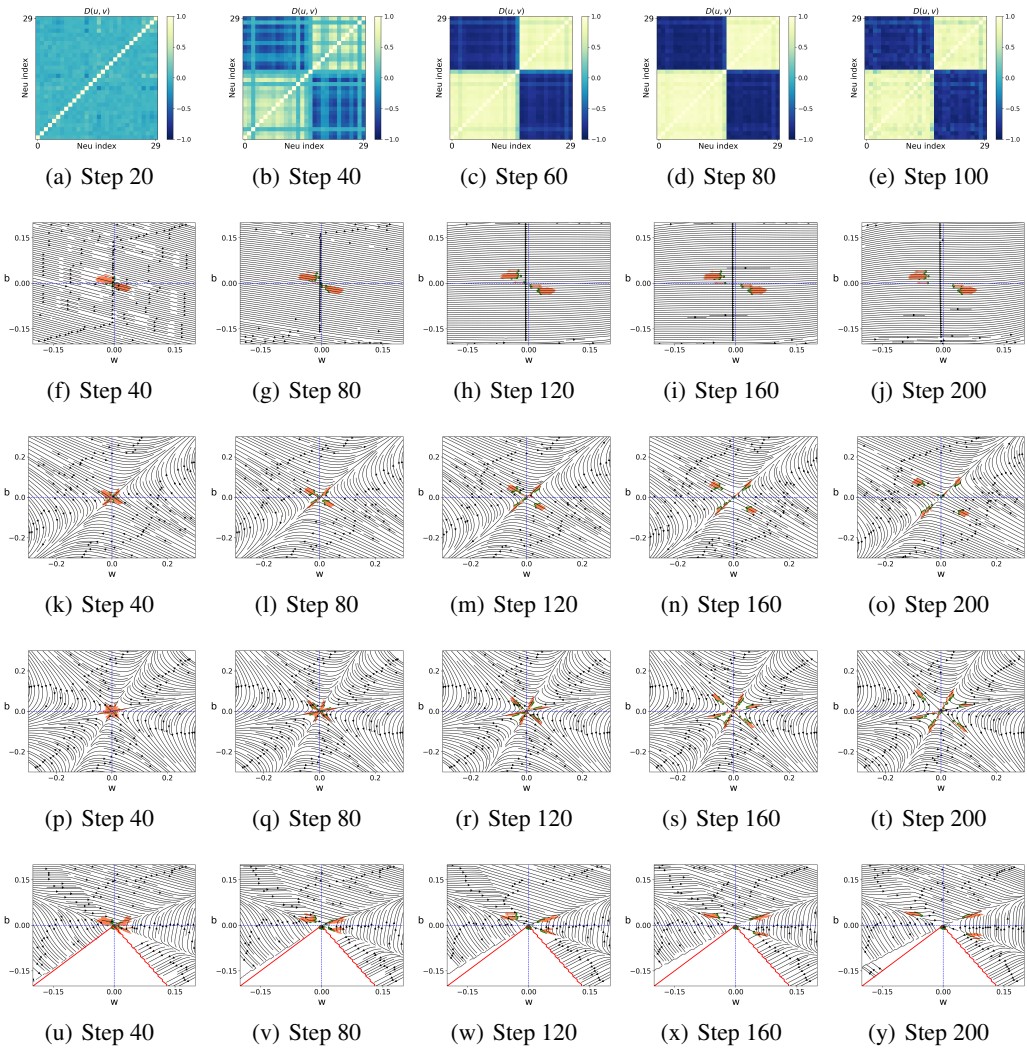

(a) Step 20    (b) Step 40    (c) Step 60    (d) Step 80    (e) Step 100

(f) Step 40    (g) Step 80    (h) Step 120    (i) Step 160    (j) Step 200

(k) Step 40    (l) Step 80    (m) Step 120    (n) Step 160    (o) Step 200

(p) Step 40    (q) Step 80    (r) Step 120    (s) Step 160    (t) Step 200

(u) Step 40    (v) Step 80    (w) Step 120    (x) Step 160    (y) Step 200

Figure 12: Evolution of condensation from Fig. 6(a) to 6(d) and 4(b). The evolution from the first row to the fifth row are corresponding to the Fig. 4(b), Fig. 6(a), Fig. 6(b), Fig. 6(c), Fig. 6(d). The numbers of evolutionary steps are shown in the sub-captions, where sub-figures in the last row are the epochs in the article.

### A.5 THE EXPERIMENTS ON CIFAR 10 AND CIFAR 100 WITH RESNET18-LIKE NEURAL NETWORK

The condensation of the Resnet18-like neural network on CIFAR10 and CIFAR100 is shown in Fig. 13 and Fig. 14, whose activation functions for fully-connected (FC) layers are $\tanh(x)$, $\mathrm{ReLU}(x)$, $\mathrm{sigmoid}(x)$, $\mathrm{softplus}(x)$ and $x\tanh(x)$, indicated by the corresponding sub-captions, respectively.

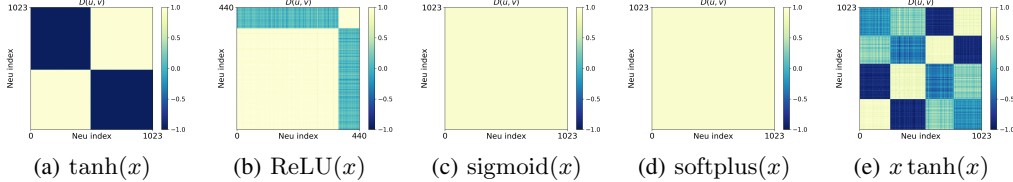

| (a) $\tanh(x)$ | (b) $\mathrm{ReLU}(x)$ | (c) $\mathrm{sigmoid}(x)$ | (d) $\mathrm{softplus}(x)$ | (e) $x\tanh(x)$ |

Figure 13: Condensation of Resnet18-like neural networks on CIFAR10. Each network consists of the convolution part of resnet18 and fully-connected (FC) layers with size 1024-1024-10 and softmax. The color in figures indicates the inner product of normalized input weights of two neurons in the first FC layer, whose indexes are indicated by the abscissa and the ordinate, respectively. We discard the hidden neurons, in which the $L_2$-norm of each input weight is smaller than 0.001, while remaining ones bigger than 0.05 in (b). The convolution part is equipped with ReLU activation and initialized by Glorot normal distribution (Glorot & Bengio, 2010). The activation functions are $\tanh(x)$, $\mathrm{ReLU}(x)$, $\mathrm{sigmoid}(x)$, $\mathrm{softplus}(x)$ and $x\tanh(x)$ for FC layers in (a), (b), (c), (d) and (e), separately. The numbers of steps selected in the sub-pictures are epoch 20, epoch 8, epoch 30, epoch 30 and epoch 61, respectively. The learning rate is $3\times10^{-8}$, $5\times10^{-6}$, $1\times10^{-8}$, $1\times10^{-8}$ and $5\times10^{-6}$, separately .The FC layers are initialized by $N(0,\frac{1}{m_{\mathrm{out}}^3})$, and Adam optimizer with cross-entropy loss and batch size 128 are used for all experiments.

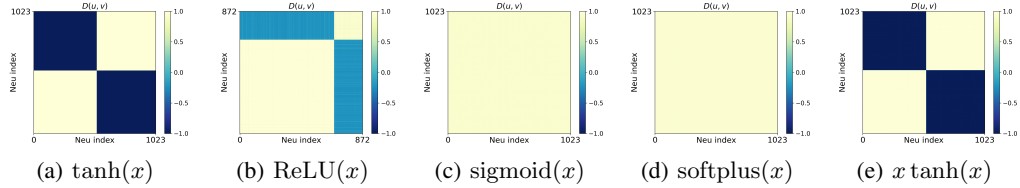

| (a) $\tanh(x)$ | (b) $\mathrm{ReLU}(x)$ | (c) $\mathrm{sigmoid}(x)$ | (d) $\mathrm{softplus}(x)$ | (e) $x\tanh(x)$ |

Figure 14: Condensation of Resnet18-like neural networks on CIFAR100. Each network consists of the convolution part of resnet18 and fully-connected (FC) layers with size 1024-1024-10 and softmax. The color in figures indicates the inner product of normalized input weights of two neurons in the first FC layer, whose indexes are indicated by the abscissa and the ordinate, respectively. We discard about 15% of the hidden neurons, in which the $L_2$-norm of each input weight is smaller than 0.0001, while remaining ones bigger than 0.0025 in (b). The convolution part is equipped with ReLU activation and initialized by Glorot normal distribution (Glorot & Bengio, 2010). The activation functions are $\tanh(x)$, $\mathrm{ReLU}(x)$, $\mathrm{sigmoid}(x)$, $\mathrm{softplus}(x)$ and $x\tanh(x)$ for FC layers in (a), (b), (c), (d) and (e), separately. The numbers of steps selected in the sub-pictures are epoch 10, epoch 10, epoch 10, epoch 10 and epoch 250, respectively. The learning rate is $1\times10^{-7}$, $1\times10^{-7}$, $3\times10^{-8}$, $3\times10^{-8}$ and $1\times10^{-6}$, separately. The FC layers are initialized by $N(0,\frac{1}{m_{\mathrm{out}}^3})$, and Adam optimizer with cross-entropy loss and batch size 128 are used for all experiments.

A.6 MULTILAYER EXPERIMENTAL

The condensation of the six layer without residual connections is shown in 15, whose activation functions for hidden layer 1 to hidden layer 5 are $x^2 \tanh(x)$, $x \tanh(x)$, $\mathrm{sigmoid}(x)$, $\tanh(x)$ and $\mathrm{softplus}(x)$, respectively.

The condensation of the three layer without residual connections is shown in 16, whose activation functions are same for each layer indicated by the corresponding sub-captions.

The condensation of the five layer without residual connections is shown in 17, whose activation functions are same for each layer indicated by the corresponding sub-captions.

The condensation of the five layer with residual connections is shown in 18, whose activation functions are same for each layer indicated by the corresponding sub-captions.

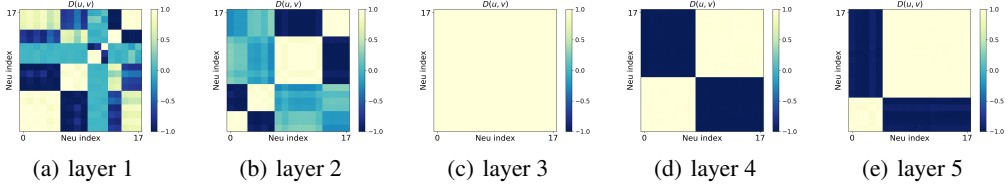

(a) layer 1     (b) layer 2     (c) layer 3     (d) layer 4     (e) layer 5

Figure 15: Condensation of six-layer NNs without residual connections. The activation functions for hidden layer 1 to hidden layer 5 are $x^2 \tanh(x)$, $x \tanh(x)$, $\mathrm{sigmoid}(x)$, $\tanh(x)$ and $\mathrm{softplus}(x)$, respectively. The numbers of steps selected in the sub-pictures are epoch 6800, epoch 6800, epoch 6800, epoch 6800 and epoch 6300, respectively, while the NN is only trained once. The color indicates $D(u,v)$ of two hidden neurons' input weights, whose indexes are indicated by the abscissa and the ordinate, respectively. The training data is 80 points sampled from a 3-dimensional function $\sum_{k=1}^{3} 4\sin(12x_k + 1)$, where each $x_k$ is uniformly sampled from $[-4, 2]$. $n = 80$, $d = 3$, $m = 18$, $d_{out} = 1$, $var = 0.008^2$, $lr = 5 \times 10^{-5}$.

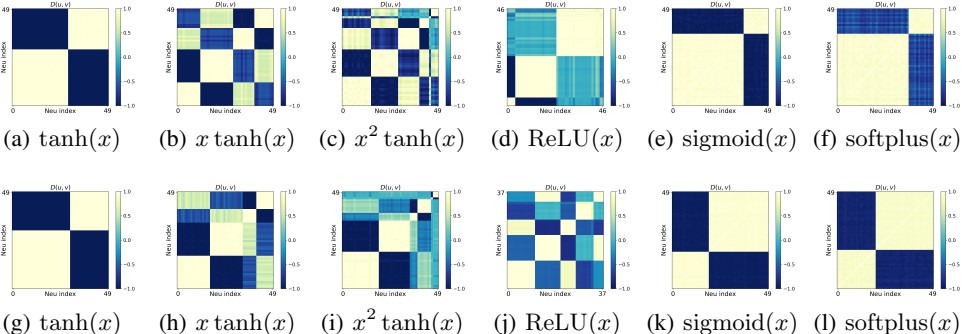

(a) $\tanh(x)$   (b) $x \tanh(x)$   (c) $x^2 \tanh(x)$   (d) $\mathrm{ReLU}(x)$   (e) $\mathrm{sigmoid}(x)$   (f) $\mathrm{softplus}(x)$

(g) $\tanh(x)$   (h) $x \tanh(x)$   (i) $x^2 \tanh(x)$   (j) $\mathrm{ReLU}(x)$   (k) $\mathrm{sigmoid}(x)$   (l) $\mathrm{softplus}(x)$

Figure 16: Three-layer NN at epoch 700. (a-f) are for the input weights of the first hidden layer and (g-l) are for the input weights of the second hidden layer. The color indicates $D(u,v)$ of two hidden neurons' input weights, whose indexes are indicated by the abscissa and the ordinate, respectively. The training data is 80 points sampled from a 5-dimensional function $\sum_{k=1}^{5} 3\sin(8x_k + 1)$, where each $x_k$ is uniformly sampled from $[-4, 2]$. $n = 80$, $d = 5$, $m = 50$, $d_{out} = 1$, $var = 0.005^2$. $lr = 10^{-4}, 2 \times 10^{-5}, 1.4 \times 10^{-5}$ for (a-d), (e) and (f), respectively. For (d) and (j), we discard hidden neurons, whose $L_2$-norm of its input weight is smaller than $0.1$.

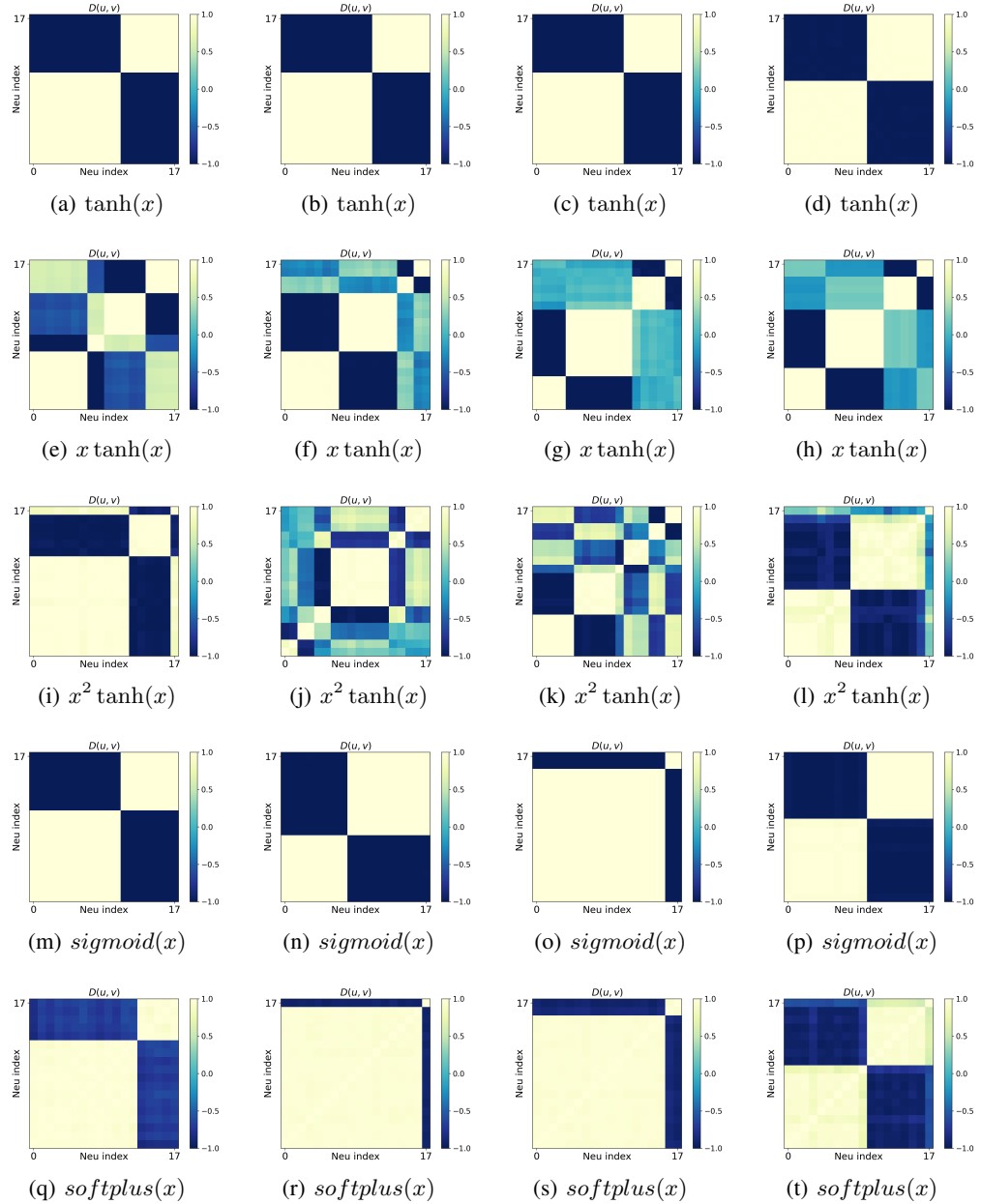

Figure 17: Five-layer NN. The first to fourth columns of each row are for the input weights of neurons from the first to the fourth hidden layers, respectively. The color indicates $D(u, v)$ of two hidden neurons' input weights, whose indexes are indicated by the abscissa and the ordinate, respectively. The training data is 80 points sampled from a 5-dimensional function $\sum_{k=1}^{3} 3\sin(10x_k + 1)$, where each $x_k$ is uniformly sampled from $[-4, 2]$. $n = 80$, $d = 5$, $m = 18$, $d_{out} = 1$, $var = 0.008^2$. $lr = 1.5 \times 10^{-5}$, $1.5 \times 10^{-5}$, $1.5 \times 10^{-5}$, $1.5 \times 10^{-5}$, $1.5 \times 10^{-6}$ and epoch is 10000, 10000, 26000, 10000, 20000 for $tanh(x)$, $xtanh(x)$, $x^2tanh(x)$, $sigmoid(x)$, $softplus(x)$, respectively.

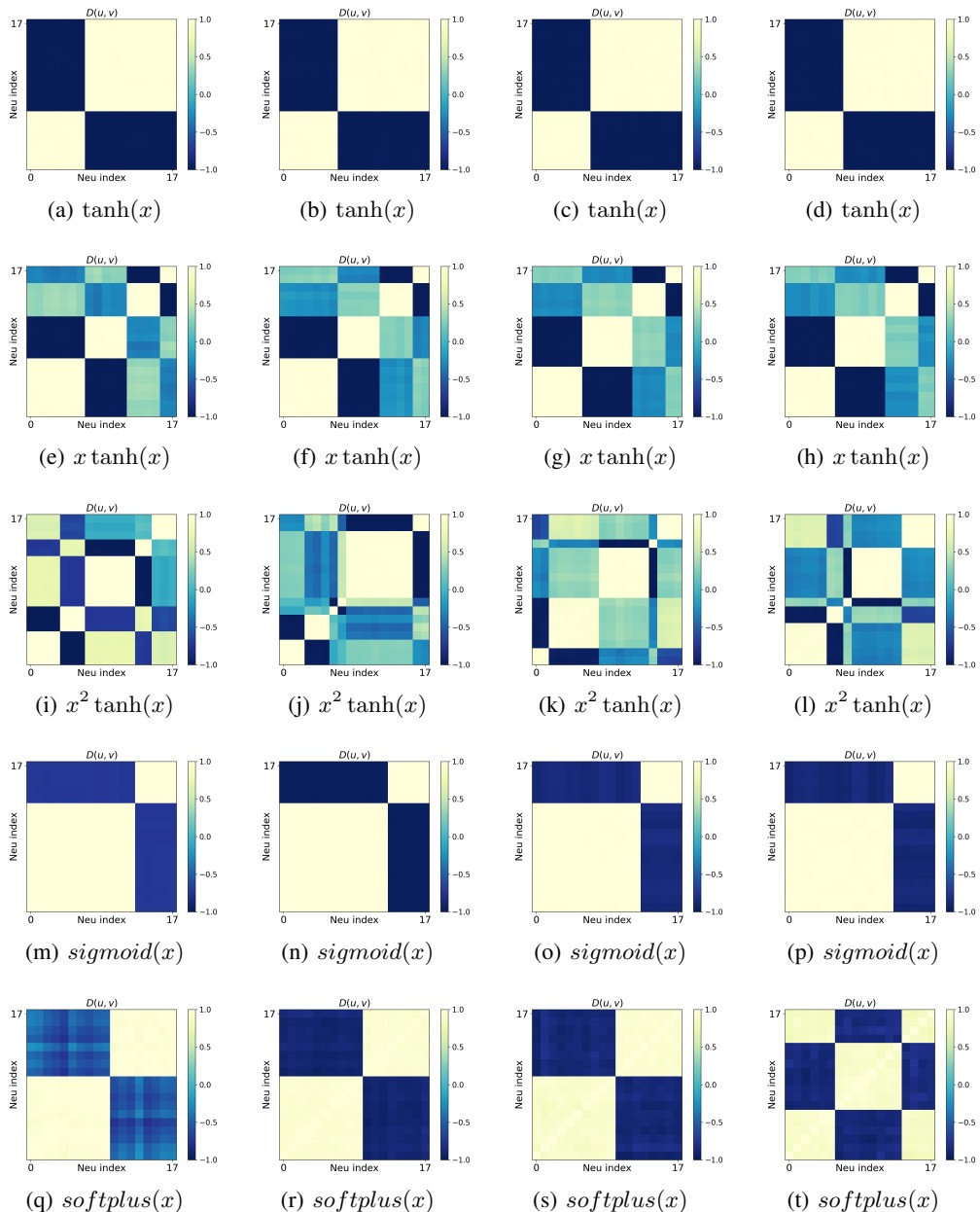

Figure 18: Five-layer NN. The first to fourth columns of each row are for the input weights of neurons from the first to the fourth hidden layers, respectively. The color indicates $D(u, v)$ of two hidden neurons' input weights, whose indexes are indicated by the abscissa and the ordinate, respectively. The training data is 80 points sampled from a 5-dimensional function $\sum_{k=1}^{3} 3\sin(10x_k + 1)$, where each $x_k$ is uniformly sampled from $[-4, 2]$. $n = 80, d = 5, m = 18, d_{out} = 1, var = 0.008^2$. $lr = 1 \times 10^{-4}, 1 \times 10^{-4}, 1 \times 10^{-4}, 5 \times 10^{-5}, 5 \times 10^{-5}$ and epoch is 400, 400, 400, 3000, 360, 400 for $tanh(x), xtanh(x), x^2 tanh(x), x^2 tanh(x), sigmoid(x), softplus(x)$, respectively.

