# OpenReview forum: "Towards Understanding the Condensation of Neural Networks at Initial Training"
_ICLR.cc/2022/Conference — ICLR 2022 Submitted_

### Official Review · Reviewer_jDJ5 · 2021-10-28

**Correctness:** 3
**Technical Novelty And Significance:** 3
**Empirical Novelty And Significance:** 2
**Recommendation:** 6
**Confidence:** 3

**Main Review:**

Strengths:
- The paper is well written, organized, and easy to follow. The experimental part is comprehensive and a detailed visualization of condensation orientations is provided.
- The conclusion that the number of condensation orientations is twice the multiplicity of the activation function gives an possible explanation why small initialization works as implicit regularization at initial training and is insightful for diving into the nonlinear dynamics of NN.

Weaknesses:
- For experiments on real datasets such as MNIST and CIFAR10, the output dimension $d_\text{out}$ is still 1. In that case, what is the target value? Is it just the class index, e.g., 0-9?
- It is hard to analyse the case of the general  multiplicity with high-dimensional input data. However, as Figure 2 displays condensation orientations for activation functions with the multiplicity>1 such as $x\tanh(x)$ and $x^2\tanh(x)$, I think the analysis of $p=2,3$ can make the conclusion more stronger.
- An extension to more practical cases where the output dimension is greater than one is interesting and needs some discussions. Also, when combined with the softmax function for a classification task, does the conclusion still hold?


**Summary Of The Paper:**

This paper investigates the condensation of weights of neural networks during the initial training stage. It showed theoretically and empirically that the maximal number of condensed orientations in the initial training stage is twice the multiplicity of the activation function under the small initialization of weights. This condensation restricts the capacity of NNs at the beginning, working as implicit regularization.

**Summary Of The Review:**

This paper presents a possible perspective to explain why small initialization can serve as implicit regularization during the initial training stage and provides insights to investigate the nonlinear dynamics of neural networks. While it can be improved by conducting more experiments as well as analysis of more complicated cases, it is above the acceptance threshold.

---

> ### Author Response · Authors · 2021-11-16
> **Response to Reviewer jDJ5**
>
> Question 1: For experiments on real datasets such as MNIST and CIFAR10, the output dimension
>  is still 1. In that case, what is the target value? Is it just the class index, e.g., 0-9?
>
> Reply 1: Our typos in previous submission causes this misunderstanding of our experiments. For CIFAR10, we use cross-entropy as loss function with output dimension $d_{out} = 10$. And for MNIST, we use MSE as loss function with output dimension $d_{out} = 10$. In the previous version, the dimension of MNIST is 10, while we had a typo of writing it as 1 in the section "Experimental setup". We also had a typo of writing loss function of CIFAR as MSE, which actually is Cross-entropy. We clarify them in the section "Experimental setup" in revision, as
>
> "For synthetic dataset and MNIST:
> Throughout this work, we use fully-connected neural network with size, $d$-$m$-$\cdots$-$m$-$d_{out}$. The input dimension $d$ is determined by the training data. The output dimension is $d_{out}=1$ for synthetic data and $d_{out}=10$ for MNIST. The number of hidden neurons $m$ is specified in each experiment. All parameters are initialized by a Gaussian distribution $N(0,var)$. The total data size is $n$. The training method is Adam with full batch, learning rate $lr$ and MSE loss.
> For synthetic data, we sample the training data uniformly from a sub-domain of $\mathbb{R}^{d}$.
>
>
> For CIFAR10 and CIFAR100 dataset:  We use Resnet18-like neural network, which has been describe in Fig. 1 thoroughly. The input dimension $d$ is determined by the training data. The output dimension is $d_{out}=10$ for CIFAR10 and $d_{out}=100$ for CIFAR100. All parameters are initialized by a Gaussian distribution $N(0,var)$. The total data size is $n$. The training method is Adam with batch size 128, learning rate $lr$ and Cross-entropy loss. "
>
> Besides, we also add several experiments on dataset CIFAR100 with Resnet18-like networks as done for CIFAR10.
>
> Question 2: It is hard to analyse the case of the general multiplicity with high-dimensional input data. However, as Figure 2 displays condensation orientations for activation functions with the multiplicity>1 such as  $xtanh$ and $x^2tanh$
> , I think the analysis of $p=2,3$ can make the conclusion more stronger.
>
> Reply 2: Firstly, for theoretical analysis, we considered the following two cases: firstly, multiplicity $p=1$, with dimension $d$ arbitrary; secondly, the dimension $d=1$, with multiplicity $p$ arbitrary. Because commonly used activation functions, such as tanh(x), sigmoid(x), softplus(x), etc., are all multiplicity one, the theoretical analysis explains the situation in most practical training.
>
> Secondly, the analysis of $p=2,3$ indeed could make the conclusion more stronger. However, the problem of p=2 or 3 is extremely difficult. When p=2/3, the problem is equivalent to finding the number of solutions in n second-order/third-order polynomial equations system, and these equations seem have a special structure generated by neural networks. Such a problem is also difficult in mathematics.
>
> Question 3: An extension to more practical cases where the output dimension is greater than one is interesting and needs some discussions. Also, when combined with the softmax function for a classification task, does the conclusion still hold?
>
> Reply 3: As mentioned before, some typos in the previous manuscript cause these misunderstandings. We have done experiments with the softmax function for a classification task and the conclusion still holds.
>
> For theoretical analysis, we do not depend on the dimension of the output, and we have also added this statement to the section "Preliminary: Neural networks and initial stage" in the revision.
>
> "Without loss of generality, we assume that the output is one-dimensional for theoretical analysis, because for high-dimensional cases, we only need to sum the components directly. For summation, it does not affect the results of our theories. "

---

> > ### Comment · Reviewer_jDJ5 · 2021-11-29
> > **Thanks for the response**
> >
> > Thanks for your detailed response. After reading other reviewers' comments and your response, I keep my original score.

---

### Official Review · Reviewer_zuZq · 2021-11-03

**Correctness:** 4
**Technical Novelty And Significance:** 3
**Empirical Novelty And Significance:** 3
**Recommendation:** 6
**Confidence:** 3

**Main Review:**

- The biggest thing that I would like is a better characterization of the experiments they run.  Currently they show multiplicity of 1 with little training and multiplicity >1 at much more training. I want the 2x2: what is the correlation matrix for the p>1 at short time scales and for p==1 at longer time scales? E.g., in Fig 1, does relu net at epoch 61 exhibit the bifurcation seen in the xtanh(x)? And is the xtanh(x) condensing on one before?
  - More broadly, it would be great to see these matrices for a few example networks at a dozen timepoints over the course of training (spanning the periods before they condense to after they condense) -- this could be an informative appendix figure
  - Also, please state what proportion of units are discarded because the l2 norm was below threshold (both in Fig 1 and in all cases)

- Is there any rigorous or quantitative sense of what the authors mean by "initial"?

- A key component of the intuitive argument and analytic results is that the weights have a very low magnitude upon initialization. It is important that the authors did indeed show that at least for CIFAR10, a network with small initialization achieves comparable performance to a more standardly initialized net and does so in a similar amount of training time. But I was left wondering: What are the implications of these results for the more standard case of larger initializations? Presumably this condensation on one or a few axes is not seen in these cases, but I'd be curious about the authors speculations about this. Do the authors have any suggestion as to whether there is some sense in which the net is effectively lower capacity in those cases?

- Why is it natural to vary the nonlinearity across layers of a single network (e.g., second paragraph on page 5)? This feels relatively unmotivated to me, as that's not a standard approach in the field.

- A minor point that is not necessary here, but would helpful in future cases: It is often clearer to use diverging colormaps for positive and negative values rather than a sequential colormap (e.g., https://matplotlib.org/stable/tutorials/colors/colormaps.html). The fact that beige was an anti-correlation, not orthogonality, took me longer to realize than it should have.

- A few small typos:
- pg 2: "quantitatively theory explanation" --> ?? "quantitative theoretical explanation"?
- pg 4: "full batch expect for..." --> "full batch *except* for..."

**Summary Of The Paper:**

This well-written paper takes a step forward in understanding the implicit regularization in neural net optimization. The authors offer empirical evidence that the complexity of the function initially learned by nets is related to the multiplicity of the activation function at zero (i.e., the number of derivatives with nonzero values when evaluated at zero). Then, analytically, they show that all input weights converge towards the same or opposite direction for a multiplicity of 1 (which is the case for common activation functions like tanh, sigmoid, softplus), and that this holds for any multiplicity in the special case of a 1-dimensional hidden layer. Broadly, this work is intriguing, but could stand to benefit from a few improvements, suggested below.

**Summary Of The Review:**

This intriguing paper demonstrates, through experiment and analysis, a connection between the implicit regularization of certain networks and the multiplicity of their activation function.

---

> ### Author Response · Authors · 2021-11-16
> **Response to Reviewer zuZq**
>
>
>
> Question 4: Also, please state what proportion of units are discarded because the l2 norm was below threshold (both in Fig 1 and in all cases)
>
> Reply 4: We add the proportion of units in our revision in Fig. 1 and Fig. 4, as
>
> "We discard about 55% of the hidden neurons, in which the $L_2$-norm of each input weight is smaller than $0.001$, while remaining ones bigger than $0.05$  in (b). "
>
> "For (a), we discard about 15% of hidden neurons, in which the $L_2$-norm of each input weight is smaller than $0.04$, while remaining those bigger than $0.4$. "
>
> Question 5:  Is there any rigorous or quantitative sense of what the authors mean by "initial"?
>
> Reply 5: In order to clarify what the initial stage is, we add the definition of the initial stage at the end of the section "Preliminary: Neural networks and initial stage" in revision and put the loss diagrams of the experiments in the Appendix A.3, that is
>
> "In the experiments, we study the condensation at the initial stage of training. For a fixed loss, the step we need to achieve it is highly related to the size of learning rate. Therefore, we propose a definition of the initial stage of training by the size of loss in this article, that is the stage before the value of loss function decays to 70% of its initial value. Such a definition is reasonable, for generally a loss could decay to 1% of its initial value or even lower. The loss of the all experiments in the article can be found in Appendix A.3, and they do meet the definition of the initial stage here."
>
> Question 6: A key component of the intuitive argument and analytic results is that the weights have a very low magnitude upon initialization. It is important that the authors did indeed show that at least for CIFAR10, a network with small initialization achieves comparable performance to a more standardly initialized net and does so in a similar amount of training time. But I was left wondering: What are the implications of these results for the more standard case of larger initializations? Presumably this condensation on one or a few axes is not seen in these cases, but I'd be curious about the authors speculations about this. Do the authors have any suggestion as to whether there is some sense in which the net is effectively lower capacity in those cases?
>
> Reply 6: As the scale of parameter initialization becomes larger, the condensation becomes weaker. The understanding from a complete condensation would benefit our understanding of the training process in common initialization, that is there is an effect, although not so strong, that can limit the complexity of the neural network in the initial training. We add this discussion in the Section "Discussion".
>
> Question 7: Why is it natural to vary the nonlinearity across layers of a single network (e.g., second paragraph on page 5)? This feels relatively unmotivated to me, as that's not a standard approach in the field.
>
> Reply 7: Firstly, we also show the results of the multi-layer neural networks with the same activation function for all layers in the Appendix A.6 in our previous submission, whose results are similar to the experiments in Fig. 3.  Secondly, the reason why we carry out experiments with different activation functions for different layers within the same network is to save the space. Because the space required to post all the experiments is relatively large, while the space for the article is limited. Therefore, in order to save space, we can only put these results in the Appendix A.6. So in order to make our results representative, we vary the non-linearity across layers of a single network.
>
> Question 8: A minor point that is not necessary here, but would helpful in future cases: It is often clearer to use diverging colormaps for positive and negative values rather than a sequential colormap (e.g., https://matplotlib.org/stable/tutorials/colors/colormaps.html). The fact that beige was an anti-correlation, not orthogonality, took me longer to realize than it should have.
>
> Reply 8: We thank you for your very helpful suggestion sincerely. Diverging colormaps for positive and negative values could indeed make it clearer to distinguish the values. We will adopt it in the follow-up work.

---

> > ### Comment · Reviewer_zuZq · 2021-11-29
> > **Response to authors**
> >
> > I appreciate the authors' response to my and others' comments, including clearer description of the experiments and better documentation of the evolution of the weights across many points of training. After reviewing their comments and the updated manuscript, my scores remain the same.

---

> ### Author Response · Authors · 2021-11-16
> **Response to Reviewer zuZq**
>
> Question 1: The biggest thing that I would like is a better characterization of the experiments they run.
>
> Reply 1: We clarify the experiment settings in the section "Experimental setup" in our revision and correct some typos, as
>
> "For synthetic dataset and MNIST:
> Throughout this work, we use fully-connected neural network with size, $d$-$m$-$\cdots$-$m$-$d_{out}$. The input dimension $d$ is determined by the training data. The output dimension is $d_{out}=1$ for synthetic data and $d_{out}=10$ for MNIST. The number of hidden neurons $m$ is specified in each experiment. All parameters are initialized by a Gaussian distribution $N(0,var)$. The total data size is $n$. The training method is Adam with full batch, learning rate $lr$ and MSE loss.
> For synthetic data, we sample the training data uniformly from a sub-domain of $\mathbb{R}^{d}$.
>
>
> For CIFAR10 and CIFAR100 dataset:  We use Resnet18-like neural network, which has been describe in Fig. 1 thoroughly. The input dimension $d$ is determined by the training data. The output dimension is $d_{out}=10$ for CIFAR10 and $d_{out}=100$ for CIFAR100. All parameters are initialized by a Gaussian distribution $N(0,var)$. The total data size is $n$. The training method is Adam with batch size 128, learning rate $lr$ and Cross-entropy loss.
> ''
>
> Moreover, in the revision, we also show the diagrams of the evolution of condensation in Appendix A.4 to better observe the phenomenon. We add the following statement at the end of the section "Initial condensation of input weights",
>
> "In figures of this section, we only show the results of the final step of condensation. To facilitate the understanding of evolution of condensation in the initial stage, we show several steps during the initial stage of each example in Appendix A.4."
>
> From the evolution of condensation, we could directly observe how the condensation occurs.
>
>
> Question 2: Currently they show multiplicity of 1 with little training and multiplicity $>$1 at much more training. I want the 2x2: what is the correlation matrix for the $p>1$ at short time scales and for p==1 at longer time scales? E.g., in Fig 1, does relu net at epoch 61 exhibit the bifurcation seen in the xtanh(x)?
>
> Reply 2:  Only the epoch number could hardly reflect the initial stage, which also depends on the learning rate. Therefore, we use the loss to indicate the initial stage. Learning rate is not sensitive to the appearance of condensation. However, a small learning rate enables us to clearly observe the condensation process in the initial stage under a gradient flow training. For example, when the learning rate is relatively small, the initial stage of training may be relatively long, while when the learning rate is relatively large, the initial stage of training may be relatively small.
>
> In order to clarify what the initial stage is, we add the definition of the initial stage at the end of the section "Preliminary: Neural networks and initial stage" in revision and put the loss diagrams of the experiments in the Appendix A.3, that is
>
> "In the experiments, we study the condensation at the initial stage of training. For a fixed loss, the step we need to achieve it is highly related to the size of learning rate. Therefore, we propose a definition of the initial stage of training by the size of loss in this article, that is the stage before the value of loss function decays to 70% of its initial value. Such a definition is reasonable, for generally a loss could decay to 1% of its initial value or even lower. The loss of the all experiments in the article can be found in Appendix A.3, and they do meet the definition of the initial stage here."
>
> We empirically find that to ensure the training process follows a gradient follow, where the loss decays monotonically, we have to select a smaller learning rate for large multiplicity $p$. Therefore, it looks like we have a longer training in our experiments with large $p$. Note that for a small learning rate in the experiments of small $p$, we can observe similar condensation.
>
> As mentioned in the previous answer, we also show the condensation process in Appendix A.4.
>
> Question 3:  And is the xtanh(x) condensing on one before? More broadly, it would be great to see these matrices for a few example networks at a dozen time points over the course of training -- this could be an informative appendix figure.
>
> Reply 3: The weights in experiments of xtanh will not condense in one line in the intial training. Following your suggestion, we show the diagrams of the evolution of condensation in Appendix A.4 to better observe this phenomenon. Our theory could also confirm this point to some extent. We revise the manuscript as follow,
>
> "In figures of this section, we only show the results of the final step of condensation. To facilitate the understanding of evolution of condensation in the initial stage, we show several steps during the initial stage of each example in Appendix A.4."

---

### Official Review · Reviewer_KucV · 2021-11-08

**Correctness:** 3
**Technical Novelty And Significance:** 2
**Empirical Novelty And Significance:** 2
**Recommendation:** 5
**Confidence:** 3

**Main Review:**

The paper aims at giving an intriguing analysis of the properties of NN activations in early stages of learning, which would motivate the intrinsic generalization abilities of deep neural architectures (e.g., when over parametrized).
The paper includes both an empirical and a theoretical analysis.
My major concern with this paper is that I found the experimental session not fully convincing. First, the analysis is limited to a set of specific condition, for which it is shown that a sort of empirical relation is in place. These assumptions should be made very clear in order to avoid overstating and confusion in readers. Second, the paper focuses on claiming to analyze the behavior of the neurons activations at initialization and early stages of learning, with small weights. In this respect, I could find that the experiments use a number of 100 (or 1000) epochs, which seems something different from initial stages of training. An evolution analysis (e.g., showing the behavior over all the epochs) would have been much more informative in this regard. Moreover, while the theoretical analysis assumes small weight values, in some plots (e.g., Fig 4) the results are shown for weight values larger than a threshold (0.4), and this is a bit confusing.

In order to better understand the novelty of this contribution I also suggest considering clarifying the differences between this current work and the work in Luo et al., 2021.

The paper presents several typos and unclear sentences in the current form.

[edit after the revision]
I would like to sincerely thank the authors for the effort in improving the manuscript. I am going to increase slightly my evaluation of the paper. Thanks.


**Summary Of The Paper:**

The paper studies the relation between condensation of neurons activations and the multiplicity of the used non-linearity. Essentially, it is found an empirical link between the multiplicity of the activation function and the number of condensation directions.

**Summary Of The Review:**

The quality of the paper is hampered by
- not fully convincing and unclear experimental analysis
- low readability due to several typos in the manuscript
- unclear improvement wrt the work in Luo et al, 2021

---

> ### Author Response · Authors · 2021-11-16
> **Response to Reviewer KucV**
>
> Question 2: Second, the paper focuses on claiming to analyze the behavior of the neurons activations at initialization and early stages of learning, with small weights. In this respect, I could find that the experiments use a number of 100 (or 1000) epochs, which seems something different from initial stages of training. An evolution analysis (e.g., showing the behavior over all the epochs) would have been much more informative in this regard.
>
> Reply 2: Learning rate is not sensitive  to the appearance of condensation. However, a small learning rate enables us to clearly observe the condensation process in the initial stage under a gradient flow training. For example, when the learning rate is relatively small, the initial stage of training may be relatively long, while when the learning rate is relatively large, the initial stage of training may be relatively small. To better clarify what is the initial stage of training, we add the definition of the initial stage at the end of the section "Preliminary: Neural networks and initial stage" in revision and show the loss diagrams of the experiments in the Appendix A.3, that is
>
> "In the experiments, we study the condensation at the initial stage of training. For a fixed loss, the step we need to achieve it is highly related to the size of learning rate. Therefore, we propose a definition of the initial stage of training by the size of loss in this article, that is the stage before the value of loss function decays to 70% of its initial value. Such a definition is reasonable, for generally a loss could decay to 1% of its initial value or even lower. The loss of the all experiments in the article can be found in Appendix A.3, and they do meet the definition of the initial stage here."
>
> Moreover, in the revision, we also show the diagrams of the evolution of condensation in Appendix A.4 to better observe the phenomenon. We add the following statement at the end of the section "Initial condensation of input weights",
>
> "In figures of this section, we only show the results of the final step of condensation. To facilitate the understanding of evolution of condensation in the initial stage, we show several steps during the initial stage of each example in Appendix A.4."
>
> From the evolution of condensation, we could directly observe how the condensation occurs.
>
> Question 3: Moreover, while the theoretical analysis assumes small weight values, in some plots (e.g., Fig 4) the results are shown for weight values larger than a threshold (0.4), and this is a bit confusing.
>
> Reply 3: The L2-norm is $||w||_{2} = \sqrt{\sum_i w_i^2}$. The number for elements in a weight is 785 (including bias). Therefore, the mean magnitude for each parameter (0.4$^2$/785)$^{0.5}$ $\sim$0.01, which should be quite small. We clarify this point in the revised manuscript.
>
> Question 4: clarifying the differences between this current work and the work in Luo et al., 2021.
>
> Reply 4: Luo et al., (2021). only report there is a condensed regime in two-layer ReLU neural network with infinite width and MSE, in which weights condense during training. In our work, we show the condense phenomenon in finite-width networks with multi-layer, general activation function, MSE loss and cross-entropy loss. In addition, we perform a theoretical analysis that includes the case of p=1 for general fully connected network with arbitrary dimension input, which is common in practice.
> We clarify the differences in the revised manuscript in Section Related works:
>
> "however, in Luo et al., (2021), it is not clear how general of the condensation when other activation functions are used and why there is condensation."

---

> ### Author Response · Authors · 2021-11-16
> **Response to Reviewer KucV**
>
> Question 1: First, the analysis is limited to a set of specific condition, for which it is shown that a sort of empirical relation is in place. These assumptions should be made very clear in order to avoid overstating and confusion in readers.
>
> Reply 1: We have stated clearly our conditions in both experiments and theory. For your reference, we recapitulate them here.
>
> For experimental analysis, we carry out experiments on: synthetic functions, MNIST and CIFAR10; fully connected neural network, convolution neural network and with residual connections; MSE and Cross-entropy; GD and SGD. To better clarify the experiment settings, we revise the section "Experimental setup" as
>
> "For synthetic dataset and MNIST:
> Throughout this work, we use fully-connected neural network with size, $d$-$m$-$\cdots$-$m$-$d_{out}$. The input dimension $d$ is determined by the training data. The output dimension is $d_{out}=1$ for synthetic data and $d_{out}=10$ for MNIST. The number of hidden neurons $m$ is specified in each experiment. All parameters are initialized by a Gaussian distribution $N(0,var)$. The total data size is $n$. The training method is Adam with full batch, learning rate $lr$ and MSE loss.
> For synthetic data, we sample the training data uniformly from a sub-domain of $\mathbb{R}^{d}$.
>
>
> For CIFAR10 and CIFAR100 dataset:  We use Resnet18-like neural network, which has been describe in Fig. 1 thoroughly. The input dimension $d$ is determined by the training data. The output dimension is $d_{out}=10$ for CIFAR10 and $d_{out}=100$ for CIFAR100. All parameters are initialized by a Gaussian distribution $N(0,var)$. The total data size is $n$. The training method is Adam with batch size 128, learning rate $lr$ and Cross-entropy loss. "
>
> For theoretical analysis, we considered the following two cases: firstly, multiplicity $p=1$, with dimension $d$ arbitrary; secondly, the dimension $d=1$, with multiplicity $p$ arbitrary. Because commonly used activation functions, such as tanh(x), sigmoid(x), softplus(x), etc., are all multiplicity one, the theoretical analysis explains the situation in most practical training. The following statements are quoted from the section "Analysis of the initial condensation of input weights",
>
> "Suppose  the activation function has multiplicity $p$, i.e., $\sigma^{(k)}(0) = 0$ for $k=1,2,\cdots, p-1$, and $\sigma^{(p)}(0) \neq 0$. For convenience, we define an operator $\mathcal{P}$ satisfying
> $\mathcal{P} \mathbf{w}:=\dot{\mathbf{w}} - \mathbf{u} (\dot{\mathbf{w}} \cdot \mathbf{u}).$
> Condensation refers to that the weight evolves  towards a direction that will not change in the direction field and is defined as follows,
> $$\dot{\mathbf{u}}=0 \ \Leftrightarrow \ \mathcal{P} \mathbf{w}:=\dot{\mathbf{w}} - \mathbf{u} (\dot{\mathbf{w}} \cdot \mathbf{u}) = 0.$$
> "
>
> "In this work, we consider NNs with sufficiently small parameters."
>
> "In the following, we study the case of (i) $p=1$ and (ii) $m_l=1$.  "

---

### Official Review · Reviewer_ejGJ · 2021-11-08

**Correctness:** 3
**Technical Novelty And Significance:** 2
**Empirical Novelty And Significance:** 2
**Recommendation:** 5
**Confidence:** 2

**Main Review:**

Strengths:
- Provide an interesting explanation of the condensation mechanism through the lens of multiplicity of activation functions

Weaknesses:
- Theoretical analysis is limited to the case of one-dimensional inputs and activation functions of multiplicity one
- Experimental demonstrations are somehow limited -- could be further improved by considering how similar phenomenon occurs for a range of other datasets and for other loss functions (cross-entropy, etc.)

Minor comments/questions:
- the focus seems to be on how condensation appears during the initial training stage, but there is no discussion of how initial this stage is (in particular, the selection of the number of epochs in the plots shown in the paper seems unjustified)
- how does the learning rate play a role in the appearance of condensation? Would be nice to at least mention this

**Summary Of The Paper:**

This paper studies the role of activation functions (via their multiplicity at the origin) in the condensation of neural networks at the initial training stage. Condensation can be viewed as a feature learning process, where the wide network can be described effectively as a narrower network and the input weights condense on isolated orientations during training. This mechanism may provide a plausible explanation for the performance of the wide network. In particular, the paper shows empirically that the maximal number of condensed orientations is twice the multiplicity of the activation function used. Moreover, using polynomial approximations, the paper provides theoretical support in two cases: when the activation function is of multiplicity one and when the input is one-dimensional.

**Summary Of The Review:**

This paper provides a study of the role of multiplicity of activation functions in the condensation mechanism of neural network training, claiming that the maximal number of condensed orientations in the initial training stage is twice the multiplicity. Some experimental demonstrations are provided to support the claim but the theory is somehow limited. I believe that the paper could be further improved and will go with a weak reject for now.

---

> ### Author Response · Authors · 2021-11-16
> **Response to Reviewer ejGJ**
>
> Question 3: the focus seems to be on how condensation appears during the initial training stage, but there is no discussion of how initial this stage is (in particular, the selection of the number of epochs in the plots shown in the paper seems unjustified).
>
>
> Reply 3: In order to clarify what the initial stage is, we add the definition of initial stage at the end of the section "Preliminary: Neural networks and initial stage" in revision and put the loss diagrams of the experiments in the Appendix A.3, that is
>
> "In the experiments, we study the condensation at the initial stage of training. For a fixed loss, the step we need to achieve it is highly related to the size of learning rate. Therefore, we propose a definition of the initial stage of training by the size of loss in this article, that is the stage before the value of loss function decays to 70% of its initial value. Such a definition is reasonable, for generally a loss could decay to 1% of its initial value or even lower. The loss of the all experiments in the article can be found in Appendix A.3, and they do meet the definition of the initial stage here."
>
> Moreover, in the revision, we also show the diagrams of the evolution of condensation in Appendix A.4 to better observe the phenomenon. We add the following statement at the end of the section "Initial condensation of input weights",
>
> "In figures of this section, we only show the results of the final step of condensation. To facilitate the understanding of evolution of condensation in the initial stage, we show several steps during the initial stage of each example in Appendix A.4."
>
> From the evolution of condensation, we could directly observe how the condensation occurs.
>
> Question 4: How does the learning rate play a role in the appearance of condensation? Would be nice to at least mention this.
>
> Reply 4: Learning rate is not sensitive to the appearance of condensation. However, a small learning rate enables us to clearly observe the condensation process in the initial stage under a gradient flow training. For example, when the learning rate is relatively small, the initial stage of training may be relatively long, while when the learning rate is relatively large, the initial stage of training may be relatively small. To better clarify what is the initial stage of training, we add the definition of the initial stage as indicated in the previous response.

---

> > ### Comment · Reviewer_ejGJ · 2021-11-29
> > **Acknowledgement of rebuttal**
> >
> > I thank the author(s) for the clarifications and revision of the paper. I am going to keep my score.

---

> > > ### Author Response · Authors · 2021-11-30
> > > **Please clarify the comment**
> > >
> > > Dear Reviewer,
> > >
> > >       Thank you for taking your time to review our rebuttal. However, you feedback is very confusing. If we have clarified your concerns, which we actually did in our revision, we hope you can raise your score. If you are still not satisfied with our revision, would you please point out? We would like to emphasize again, our revision has responded to all your concerns one point by another.
> > >
> > > Thanks!
> > >
> > > Authors

---

> ### Author Response · Authors · 2021-11-16
> **Response to Reviewer ejGJ**
>
>
> Question 1: Theoretical analysis is limited to the case of one-dimensional inputs and activation functions of multiplicity one.
>
> Reply 1: For theoretical analysis, we consider the following two cases: firstly, multiplicity $p=1$, with dimension $d$ arbitrary; secondly, the dimension $d=1$, with multiplicity $p$ arbitrary. Because commonly used activation functions, such as tanh(x), sigmoid(x), softplus(x), etc., are all multiplicity one, the theoretical analysis explains the situation in most practical training. As well-known, the theory in deep learning is difficult. For a conference paper, with the focus of empirical study, it is reasonable to claim enough analysis. To clarify our theoretical study, we revise the manuscript through out the paper,
>
> "Our theoretical analysis confirms experiments for two cases, one is for the activation function of multiplicity one with arbitrary dimension input, which contains many common activation functions, and the other is for the layer with one-dimensional input and arbitrary multiplicity. "
>
> Question 2: Experimental demonstrations are somehow limited -- could be further improved by considering how similar phenomenon occurs for a range of other datasets and for other loss functions (cross-entropy, etc.).
>
> Reply 2: We perform experiments on synthetic data, MNIST and CIFAR10 with both MSE and Cross-entropy loss, GD and SGD, which should be adequate for basic research. There are several typos in the section "Experimental setup", which may lead to a misunderstanding of our experiments. For CIFAR10, we use cross-entropy as loss function with output dimension $d_{out} = 10$. And for MNIST, we use MSE as loss function with output dimension $d_{out} = 10$. In the previous version, the dimension of MNIST is 10, while we had a typo of writing it as 1 in the section "Experimental setup". We also had a typo of writing loss function of CIFAR as MSE, which actually is Cross-entropy. We clarify them in the section "Experimental setup" in revision, as
>
> "For synthetic dataset and MNIST:
> Throughout this work, we use fully-connected neural network with size, $d$-$m$-$\cdots$-$m$-$d_{out}$. The input dimension $d$ is determined by the training data. The output dimension is $d_{out}=1$ for synthetic data and $d_{out}=10$ for MNIST. The number of hidden neurons $m$ is specified in each experiment. All parameters are initialized by a Gaussian distribution $N(0,var)$. The total data size is $n$. The training method is Adam with full batch, learning rate $lr$ and MSE loss.
> For synthetic data, we sample the training data uniformly from a sub-domain of $\mathbb{R}^{d}$.
>
>
> For CIFAR10 and CIFAR100 dataset:  We use Resnet18-like neural network, which has been describe in Fig. 1 thoroughly. The input dimension $d$ is determined by the training data. The output dimension is $d_{out}=10$ for CIFAR10 and $d_{out}=100$ for CIFAR100. All parameters are initialized by a Gaussian distribution $N(0,var)$. The total data size is $n$. The training method is Adam with batch size 128, learning rate $lr$ and Cross-entropy loss.
> "
>
> Besides, we also add several experiments on dataset CIFAR100 with Resnet18-like networks as done for CIFAR10. It may be difficult to require researchers in university to perform large dataset and large networks.

---

### Author Response · Authors · 2021-11-17
**Response to all Reviewers**

Dear reviewers,

We thank the reviewers for your thoughtful and insightful comments. We have addressed every comment, and believe that, taken together, the reviewers' comments have improved the manuscript significantly. To address reviewers' common concerns, we have added the definition of the initial stage and evolution of condense,  revised several details and improved the writings. We hope that the revised manuscript now satisfies the reviewers' requirements, which could be found at the revision PDF.

Sincerely yours,

Authors.

---

### Decision · Program_Chairs · 2022-01-20

**Decision:**

Reject

**Comment:**

*Summary:* Study isolated orientations of weights for networks with small initialization depending on multiplicity of activation functions.

*Strengths:*
- Interesting analysis of properties in early stages depending on activations.

*Weaknesses:*
- Reviewers found the settings limited.
- Reviewers found experiments limited.

*Discussion:*

In response to ejGJ authors reiterate scope of covered cases and submit to consideration that their experiments should be adequate for basic research. Reviewer acknowledges the response, but maintains their assessment (limited scope of theory, limited experiments). KucV found the experimental part limited in scope, the settings unclear (notion of early stage, compatibility with theory), and review of previous works lacking. KucV’s sincerely acknowledged authors for their efforts to address their comments and improving the manuscript, and raised their score, but maintained the experimental analysis is not fully convincing and unclear, and the comparison with prior work insufficient. zuZq also expressed concerns with the experiments and the notions and settings under consideration. They also raised questions about the comparison with standard initialization. Authors made efforts to address zuZq concerns. zuZq acknowledged this but maintained initial position that the article is just marginally above threshold. jDJ5 found the paper well written and the conclusion insightful. However, also raised concerns about the experiments the settings under consideration. Authors made efforts to address jDJ5’s concerns, who appreciated this but was not convinced to raise their score.

*Conclusion:*
Two reviewers consider this article marginally above and two more marginally below the acceptance threshold. I find the article draws an interesting connection pertaining an interesting topic. However, the reviews and discussion conclude that the article lacks in several regards that in my view still could and should be improved. Therefore I am recommending reject at this time. I encourage the authors to revise and resubmit.